# Mena regulates nesprin-2 to control actin–nuclear lamina associations, trans-nuclear membrane signalling and gene expression

Frederic Li Mow Chee [1], Bruno Beernaert [1,10], Billie G. C. Griffith[1], Alexander E. P. Loftus[1], Yatendra Kumar[2], Jimi C. Wills[1], Martin Lee [1], Jessica Valli [3], Ann P. Wheeler[4], J. Douglas Armstrong[5], Maddy Parsons [6], Irene M. Leigh[7,8], Charlotte M. Proby [7], Alex von Kriegsheim [1], Wendy A. Bickmore [2], Margaret C. Frame [1] & Adam Byron [1,9] ✉

Interactions between cells and the extracellular matrix, mediated by integrin adhesion complexes, play key roles in fundamental cellular processes, including the sensing and transduction of mechanical cues. Here, we investigate systems-level changes in the integrin adhesome in patient-derived cutaneous squamous cell carcinoma cells and identify the actin regulatory protein Mena as a key node in the adhesion complex network. Mena is connected within a subnetwork of actin-binding proteins to the LINC complex component nesprin-2, with which it interacts and co-localises at the nuclear envelope. Moreover, Mena potentiates the interactions of nesprin-2 with the actin cytoskeleton and the nuclear lamina. CRISPR-mediated Mena depletion causes altered nuclear morphology, reduces tyrosine phosphorylation of the nuclear membrane protein emerin and downregulates expression of the immuno-modulatory gene *PTX3* via the recruitment of its enhancer to the nuclear periphery. We uncover an unexpected role for Mena at the nuclear membrane, where it controls nuclear architecture, chromatin repositioning and gene expression. Our findings identify an adhesion protein that regulates gene transcription via direct signalling across the nuclear envelope.

Interactions between cells and the extracellular matrix (ECM) are mediated by transmembrane adhesion receptors, such as integrins, which associate with intracellular scaffolding, signalling and cytoskeletal proteins in adhesion complexes[1,2]. These multiprotein complexes function at focal adhesions to initiate and modulate adhesion signalling pathways and to control a range of cellular processes including cell migration, proliferation and differentiation[3,4]. Adhesion complexes are spatiotemporally regulated by biomechanical and biochemical cues in the surrounding microenvironment, acting as mechanosensory modules that physically and functionally link the ECM and the acto-myosin cytoskeleton[2,5].

The set of proteins that associate with integrin adhesion complexes (IACs) is termed the integrin adhesome, which serves as a conceptual starting point for the systems-level analysis of adhesion proteins. A literature-curated adhesome database, compiled from multiple studies that used various cell types, analytical techniques and experimental settings, documents 232 adhesion-related proteins[6,7]. The empirical characterisation of adhesion-site composition has been

enabled by the development of methods for the biochemical isolation and mass spectrometric quantification of IACs[8–13] and has revealed an unanticipated molecular complexity and diversity of IAC proteomes[14–19]. Integration of multiple fibronectin-induced IAC proteomes in silico enabled the construction of an experimentally defined meta-adhesome database, consisting of 2412 proteins[20]. A subset of the meta-adhesome, termed the consensus adhesome, contains 60 core adhesion proteins that are frequently present in IAC proteomes[20].

The consensus adhesome is enriched in proteins with classical adhesion-related functions and those containing actin-binding domains; these can be divided into four interconnected axes characterised by focal adhesion kinase (FAK)–paxillin, talin–vinculin, integrin-linked kinase–kindlin and α-actinin–zyxin–vasodilator-stimulated phosphoprotein (VASP) modules, forming integrin–actin structural connections[20,21]. In addition to well-understood functions at focal adhesions, studies have described the localisation of certain actin-binding and actin regulatory adhesion proteins distal to IACs, such as in or around the nucleus of tumour cells. For example, α-actinin, migfilin and zyxin, which have roles in the integrin–actin structural connection, have been reported to localise to the nucleus in tumour cells[22–25], while Arp2/3 and the actin-bundling protein fascin have been described at the perinuclear region during confined cell migration[26,27]. There are likely multiple levels and mechanisms of dysregulation of integrin adhesomes in pathophysiological cell states, including altered expression[28,29] and potentially altered subcellular localisation of adhesome components[22,30–34], and these remain to be fully defined. An outstanding question in cell biology is the function of adhesion and actin-binding adhesome components in the nucleus, or proximal to the nuclear membrane.

Mechanical stress is transmitted from the plasma membrane via the cytoskeleton to the linker of nucleoskeleton and cytoskeleton (LINC) complex at the nuclear envelope[35]. The LINC complex consists of the inner nuclear membrane proteins SUN1 and SUN2, the LAP2–emerin–MAN1 (LEM) domain protein emerin and inositol 1,4,5-triphosphate receptor-associated 2, also known as lymphoid-restricted membrane protein, and the outer nuclear membrane nesprin family proteins, nesprin-1, -2, -3 and -4 and KASH5. Nesprins anchor the cytoskeleton to the nuclear membrane either by direct interaction with the cytoskeleton or via accessory proteins[35,36]. For example, nesprin-3 interacts with intermediate filaments via plectin[37], while certain actin-binding proteins, such as FH1/FH2 domain-containing protein 1 (FHOD1), enhance the interaction between nesprin-2 and actin filaments[38]. In many cases, cells require an intact LINC complex with effective connection to the cytoskeleton to efficiently transmit mechanical stress to the nucleus to elicit nuclear mechanoresponses, such as nuclear deformation and repositioning, chromatin and gene repositioning, histone modification and nuclear protein conformational change and post-translational modifications, such as phosphorylation[36,39,40]. In many tumour cells, for example, actin and actin-associated proteins are strongly dysregulated and accumulate in or around the nucleus[41,42].

In this study, we interrogated the changes in the fibronectin-induced adhesome in cancer cells using patient-derived malignant keratinocytes from cutaneous squamous cell carcinoma (cSCC). We found that the human Enabled homologue, Mena (encoded by *ENAH*; also known as hMena), an actin regulatory protein of the Ena/VASP family, is enriched in the adhesome of a cSCC cell line. Our proteomic analysis connected Mena to the LINC complex component nesprin-2, and we uncovered a role for Mena at the perinuclear region of malignant keratinocytes. We found that Mena localises immediately adjacent to nesprin-2, and interacts with its C-terminal spectrin repeats (SRs), at the nuclear membrane. CRISPR/Cas9-mediated Mena depletion causes reduced nesprin-2 interactions with actin and lamin A/C, and alters nuclear morphology. Moreover, Mena loss results in reduced tyrosine phosphorylation of the nuclear membrane protein

emerin and regulates *PTX3* (which encodes pentraxin-3 (PTX3), an immunomodulatory component of the complement system) via the recruitment of its putative enhancer region to the nuclear lamina. This study elucidates a connection between the adhesome component Mena and the LINC complex by which Mena regulates actin–nuclear lamina interactions, nuclear architecture, chromatin organisation and transcription of cell migration and immune response genes. We thus report an adhesion protein that can alter gene expression by means of direct signalling across the nuclear envelope.

## Results

### Characterisation of the adhesome of cSCC

To define the adhesome of cSCC, we biochemically isolated IACs from malignant keratinocytes derived from human cSCC. Two different cSCC cell lines (from two points in the clinical progression of the disease[43]), Met1 and Met4, were seeded onto fibronectin-coated dishes to induce formation of IACs, which were stabilised using a cell-permeable crosslinker, and cell bodies were removed by detergent extraction and application of hydrodynamic force to yield IAC fractions (Fig. 1a). Western blotting confirmed enrichment of focal adhesion proteins in fibronectin-induced IACs from the cSCC cells (Fig. 1b). αV integrin (a subunit of several fibronectin-binding integrin heterodimers) and the well-characterised adhesion protein focal adhesion kinase (FAK), and its fibronectin-induced tyrosine-phosphorylated active form (FAK pY397), were present in IAC fractions, whereas proteins not typically associated with integrin-based adhesions, such as GAPDH (cytoplasmic) and histone H3 (nuclear), were not detected (Fig. 1b), indicating we had achieved effective isolation of adhesion protein complexes from patient-derived cSCC cells.

We used quantitative proteomics to characterise the composition of cSCC IACs. Liquid chromatography-coupled tandem mass spectrometry (LC-MS/MS) analysis identified 191,683 peptide–spectrum matches, from which 1727 protein groups were quantified in at least two out of three biological replicate experiments (false discovery rate (FDR) < 1%) (Supplementary Data 1). Replicate experiments were well correlated (Pearson correlation coefficient ≥ 0.966, $P < 1 \times 10^{-307}$; Supplementary Fig. 1a), and the number of proteins identified was of a similar magnitude to IAC proteomes reported for human cell lines previously[16,44]. Integrin signalling was the most over-represented functional pathway associated with the cSCC IAC subproteome (Supplementary Fig. 1b), and of the 1727 identified cSCC IAC proteins, 1024 proteins were members of the meta-adhesome[20] and 66 were described in the literature-curated adhesome[7] (Fig. 1c). We extracted the intersection set of 56 cSCC IAC proteins reported in both the meta-adhesome and the literature-curated adhesome, reasoning that these adhesion proteins, which have been determined by various experimental methods from multiple cell lines[7] in addition to proteomic analysis of isolated IACs, constitute a robust core subset of the cSCC adhesome (Supplementary Fig. 1c, Supplementary Data 2). These core cSCC adhesion proteins were significantly enriched for proteins associated with actin binding when compared to all proteins reported in both the meta-adhesome and the literature-curated adhesome (Fig. 1d).

IACs from Met4 cells were quantitatively distinct from those from Met1 cells (Supplementary Fig. 1d), with 402 proteins (23.3%) differentially enriched by at least two-fold (238 proteins increased in Met4 IACs, 164 proteins decreased in Met4 IACs; q < 0.05) (Supplementary Data 1). Of the differentially enriched IAC proteins, 10 proteins (2.49%) were core cSCC adhesion proteins (17.9% of the 56 identified core cSCC adhesion proteins). As actin binding functions were over-represented in the core cSCC adhesome (Fig. 1d), and the actin network is of fundamental importance for the functioning of focal adhesions and the transmission of tensile forces, we further examined the actin-associated proteins recruited to cSCC IACs. The actin-binding proteins talin-1 and microtubule–actin crosslinking

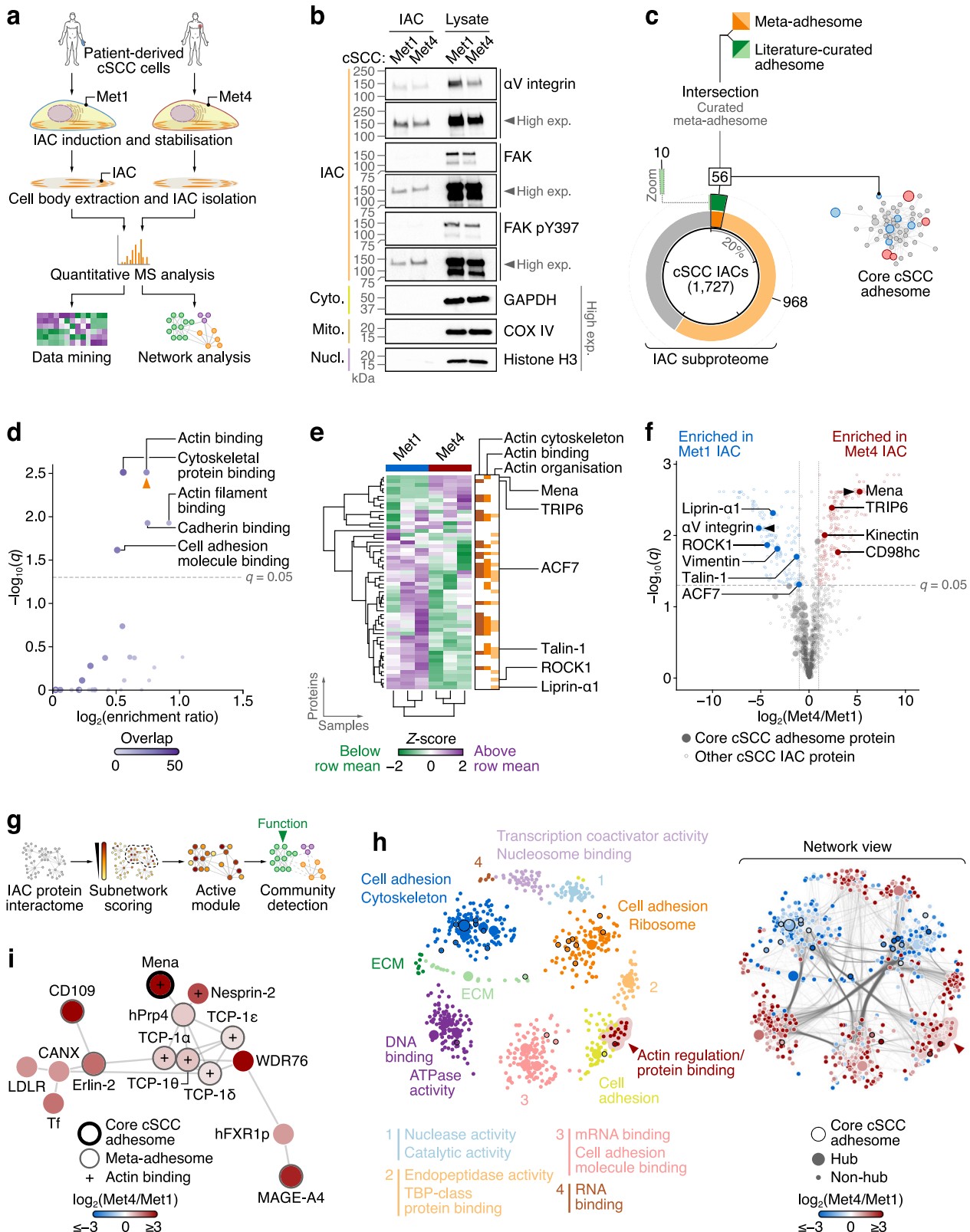

factor 1 (also known as ACF7) and the actin regulatory proteins Mena and transcriptional coactivator TRIP6, for example, were significantly differentially enriched in cSCC IACs (Fig. 1e, f; Supplementary Fig. 1e; Supplementary Data 2). Indeed, Mena was the most enriched core adhesome protein in Met4 IACs, and the high abundance of Mena in Met4 IAC fractions was confirmed by western blotting (Supplementary Fig. 1f).

To investigate Mena and its molecular connections in Met4 cells, we used graph-based analysis to assess the topology of the cSCC adhesome protein network in silico. Complex protein interaction networks can be subdivided into densely connected communities, or modules[45], which have been postulated to have functional roles in various physiological processes[46,47] and particular disease phenotypes[48,49]. We hence analysed network modularity to identify cSCC IAC proteins

**Fig. 1 | Characterisation of a patient-derived cSCC adhesome. a** Workflow for isolation and proteomic analysis of IACs. **b** Enrichment of focal adhesion proteins in cSCC IACs compared to cytoplasmic (cyto.), mitochondrial (mito.) and nuclear (nucl.) non-adhesion proteins as determined by western blotting. Images are representative of three independent experiments. Exp., exposure. **c** Proportion of the 1727 IAC proteins identified by proteomics annotated in the meta-adhesome (orange segments) or the literature-curated adhesome (green segments). IAC proteins in both the meta-adhesome and the literature-curated adhesome (inter-section set; 56 proteins; dark segments) represent a core cSCC adhesome (see Supplementary Fig. 1c for corresponding network). Inset segment (dotted box) shows zoom of literature-curated adhesome-only segment for clarity. **d** Gene Ontology over-representation analysis of molecular functions in the core cSCC adhesome. Orange arrowhead, representative functional category determined using affinity propagation. **e** Hierarchical cluster analysis of the core cSCC adhesome. Proteins associated with actin components (actin cytoskeleton), functions (actin binding) or processes (actin organisation) are indicated (orange bars). Actin-associated proteins significantly differentially abundant between Met1 and Met4

IACs are labelled ($q < 0.05$, two-sided $t$-test with Benjamini–Hochberg correction; $n = 3$ independent biological replicates). **f** Volcano plot of cSCC IAC proteins. Core adhesome proteins significantly differentially abundant between Met1 and Met4 IACs and enriched by at least two-fold are labelled; the most enriched in Met1 and Met4 IACs, respectively, are indicated with black arrowheads. **g** Workflow for graph-based analysis of the cSCC IAC subproteome. **h** Maximal-scoring active module of the cSCC IAC protein interactome. The network was partitioned using the Louvain modularity maximisation method. Proteins (nodes) are coloured according to assigned cluster (left panel). Network view (right panel) shows corresponding protein interactions (edge densities) in the partitioned network; nodes are coloured according to protein enrichment in Met1 or Met4 IACs. Black node borders indicate core adhesome proteins. The actin regulation cluster detailed in (**i**) is indicated with a red arrowhead. **i** Subnetwork analysis of the actin regulation cluster identified by active module partitioning in (**h**). Artwork in (**a**) was adapted from Byron, A., Griffith, B.G.C., Herrero, A. et al. Characterisation of a nucleo-adhesome. Nat Commun 13, 3053 (2022). https://doi.org/10.1038/s41467-022-30556-5.

that may interact with each other to form modules. An active module of a protein interaction network, also known as a responsive subnetwork, is a well-connected subnetwork enriched in proteins that change in abundance between different conditions[45]. Therefore, we reasoned that interrogating Mena in the context of an active module of the cSCC adhesome may reveal novel functional associations.

We first constructed a protein interaction network of the cSCC adhesome using curated physical protein-protein interactions from the Biological General Repository for Interaction Datasets (BioGRID) database. This enabled us to model as a graph the putative protein interactions of the cSCC IAC subproteome identified by LC-MS/MS. We integrated this graph with profiles of relative protein enrichment in Met1 and Met4 IACs quantified by MS, and for each protein (node) in the network, we computed an FDR-corrected score reflecting the significance of its differential enrichment (Fig. 1g). The maximal-scoring subnetwork was calculated using an exact approach incorporating integer linear programming[50] and extracted from the interaction network for further analysis. This high-confidence active module of the cSCC adhesome contained highly connected regions of the interaction network that show protein enrichment in Met1 or Met4 IACs (Fig. 1h). We thus captured an adhesome network hotspot associated with IAC protein recruitment in two cSCC cell lines.

We next analysed the topology of the maximal-scoring active module to infer putative functional clusters that may be regulated in the cSCC adhesome. We clustered the active module subnetwork using community detection methods and annotated the detected clusters of proteins with over-represented molecular functions, which we visualised alongside a clustered network view of corresponding protein interactions and relative protein enrichment (Supplementary Fig. 1g). The majority of clusters (7/12 identified by Louvain modularity maximisation) were annotated with cell adhesion- or cytoskeleton-related functions, with the remainder annotated with non-adhesion functions that have been linked to IAC proteins, such as transcriptional coactivator activity[51] and RNA binding[52] (Fig. 1h). This indicates that functional clusters relevant to cell adhesion proteins were identified in the cSCC adhesome network. By two independent network clustering methods, we found that Mena clustered with meta-adhesome proteins involved in filamentous actin (F-actin) binding and regulation, such as members of the chaperonin-containing TCP1 complex (CCT/TRiC)[53] (Fig. 1h, i; Supplementary Fig. 1h, i). The proteins in this cluster were all enriched in Met4 IACs (red nodes; Fig. 1h, i; Supplementary Fig. 1h, i), suggesting that they may be co-regulated and function together. In addition, Mena was proximal in the subnetwork to nesprin-2 (encoded by *SYNE2*), an F-actin-binding protein of the LINC complex (Fig. 1i; Supplementary Fig. 1i). Although nesprin-2 is not included in the current meta-adhesome database, a calponin homology domain-containing variant of nesprin-2 has been reported to localise to focal

adhesions in U2OS cells[54], and other members of the LINC complex – nesprin-1, nesprin-3, SUN1 and SUN2 – are components of the meta-adhesome[20]. Mena and nesprin-2 thus formed part of a functional cluster that was enriched in Met4 IACs, which led us to hypothesise that Mena and nesprin-2 may operate together in cSCC cells.

## Mena and nesprin-2 form a complex and co-localise at the nuclear membrane

To test the functional relevance of the network co-clustering of Mena and nesprin-2, we first examined their localisation by immuno-fluorescence imaging of Met4 cells. Spinning-disk confocal images of cells stained with anti-Mena and anti-nesprin-2 antibodies showed that, while Mena was found at a number of subcellular locales, including focal adhesions (Fig. 2a, orange arrowheads), it specifically co-localised with nesprin-2 at the nuclear membrane (Fig. 2a, magenta arrowheads). Orthogonal reconstruction of images acquired by super-resolution structured-illumination microscopy (SIM) implied that Mena abutted and partially overlapped with nesprin-2 at the nuclear membrane (Fig. 2c, green arrowheads), co-localising at the apical-most zones of nesprin-2 staining at the apical nuclear membrane, where actin filaments also co-localised (Fig. 2c, black arrowheads). Super-resolution *xz* scans acquired by 3D stimulated emission depletion (STED) microscopy revealed that Mena localisation at the nuclear membrane, where nesprin-2 is located, was distinct from the plasma membrane, whereas F-actin appeared to associate strongly with both membrane structures (Fig. 2d). Moreover, Mena co-immunoprecipitated with nesprin-2 in Met4 cell lysates (Fig. 2e). We observed multiple reactive bands to nesprin-2 by western blotting (Fig. 2e), which we inferred are likely different nesprin-2 isoforms that have been described previously[55]. Our data indicate that Mena forms a complex with nesprin-2 and that this is likely at the nuclear membrane.

It has been shown previously that SRs (spectrin repeats) of nesprin-2 are binding sites for actin-binding proteins, namely fascin[26], FHOD1 (ref. [38]) and Myc box-dependent-interacting protein 1 (also known as amphiphysin 2 or bridging integrator 1)[56]. SRs located at the C-terminal domain of nesprin-2 are the most likely sites of conserved protein-protein interaction[57,58]. We therefore tested whether Mena interacts with nesprin-2 at the C-terminal SRs by transiently trans-fecting Met4 cells with a GFP-tagged truncated nesprin-2 expressing only the eight C-terminal SRs 49–56 (GFP-nesprin-2G(SR49–56); based on the nesprin-2 giant isoform, nesprin-2G) (schematic representation in Fig. 2f). Isolation of associated protein complexes using GFP-Trap showed that Mena was highly enriched in GFP-nesprin-2G(SR49–56) pulldowns (Fig. 2g), indicating that Mena forms a complex with nesprin-2 C-terminal SRs. To further define the Mena binding site on nesprin-2, Met4 cells were transiently transfected with GFP-mini-nesprin-2GΔSR3–50, which contains SRs 51–56 of nesprin-2 (Fig. 2f),

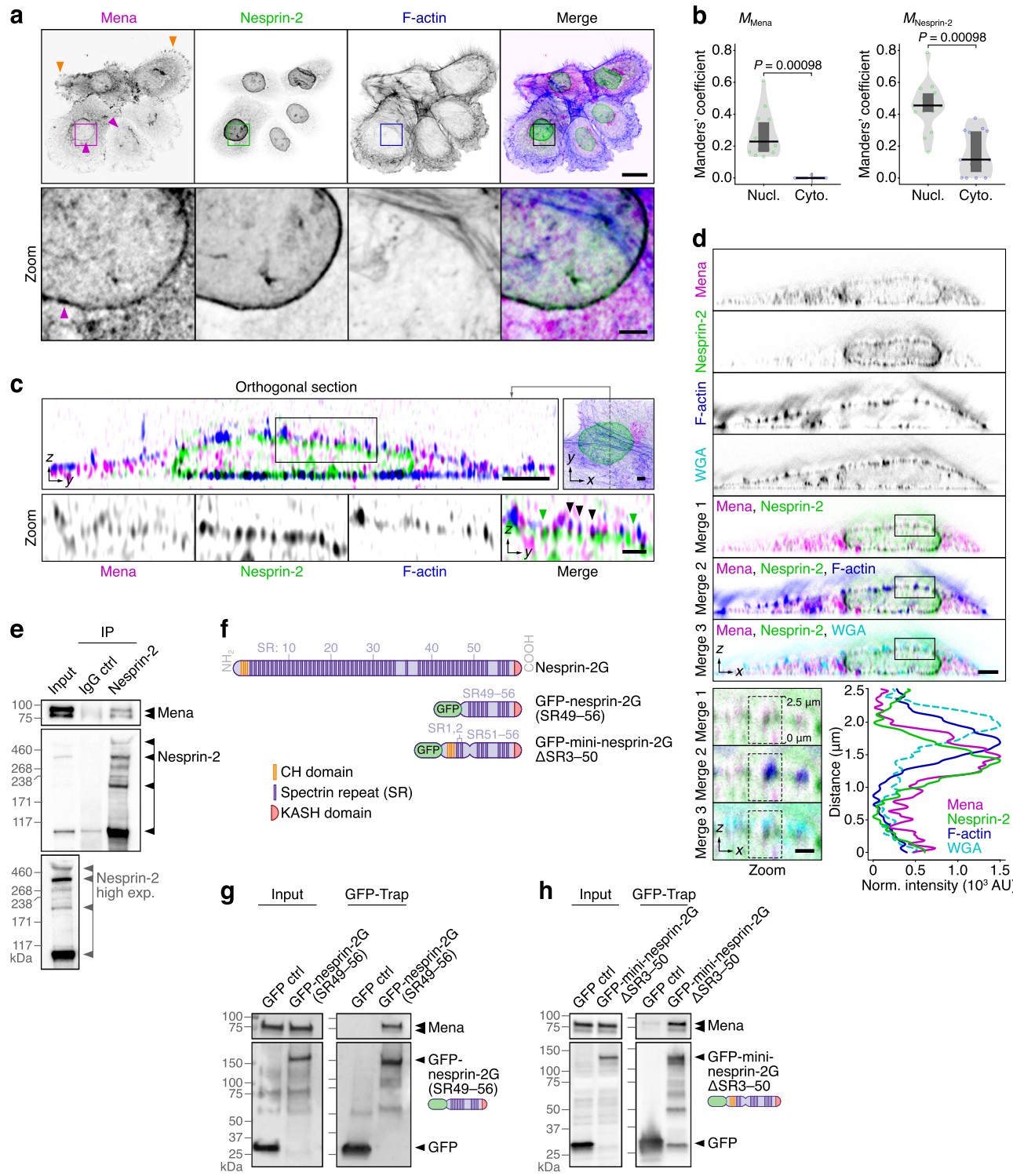

and GFP-bound protein complexes were isolated. Mena was highly enriched in GFP-mini-nesprin-2GΔSR3–50 pulldowns (Fig. 2h), implying that Mena interacts with the region of nesprin-2 containing SRs 51–56. Taken together, we conclude that Mena binds to nesprin-2 via its C-terminal SRs, and microscopy implies this occurs at the nuclear membrane where the two proteins are juxtaposed and co-localise.

## Mena depletion causes nesprin-2 dissociation from actin and lamin A/C

We next wanted to test the functional relevance of the interaction between Mena and the nesprin-2 component of the LINC complex.

Nesprin-2 interacts with actin, thereby coupling the actin cytoskeleton to the nuclear lamina via the LINC complex[35]. The LINC complex also mediates a mechanosensitive response upon physical strain at the nuclear periphery, whereby lamin A/C is recruited to the LINC complex[40]. Therefore, we examined whether the roles of nesprin-2 in binding the actin cytoskeleton and the nuclear lamina were regulated by Mena. We generated Mena-knockout Met4 cells (Met4 Mena[−/−]) using CRISPR/Cas9 and re-expressed the human Mena isoform variant 11a (Mena11a) in Met4 Mena[−/−] cells (Supplementary Fig. 2a). The Mena11a variant has an extra 63-nucleotide exon between exons 11 and 12, conferring an additional 21 amino acids to the Mena EVH2 domain[59].

**Fig. 2 | Mena co-localises with nesprin-2 at the nuclear membrane and forms a biochemical complex. a** Confocal imaging of Met4 cSCC cells. Orange arrowheads indicate Mena staining at focal adhesions; magenta arrowheads indicate Mena localisation at the nuclear membrane. Images are representative of three independent experiments. Scale bar, 20 μm; zoom (inset) scale bar, 3 μm. **b** Subcellular co-occurrence analysis of Mena and nesprin-2 signals determined from confocal images. $M_{Mena}$, co-occurrence fraction of Mena with nesprin-2 at the nuclear membrane (nucl.) or cytoplasm (cyto.). $M_{Nesprin-2}$, co-occurrence fraction of nesprin-2 with Mena at the nuclear membrane or cytoplasm. Black bar, median; dark grey box, 95% confidence interval; light grey silhouette, probability density. Statistical analysis, two-sided Wilcoxon signed rank test ($n = 14$ images from $n = 3$ independent experiments). **c** SIM imaging of Met4 cells. Position of orthogonal section (yz) shown in top-right image (xy; grey dashed line). Green arrowheads indicate co-localisation of Mena and nesprin-2 at the nuclear membrane; black arrowheads indicate co-localisation of Mena, nesprin-2 and F-actin at the nuclear membrane. Images are representative of three independent experiments. Scale bar, 3 μm; zoom (inset) scale bar, 1 μm. **d** STED imaging of Met4 cells. Signals in a region beneath the apical plasma membrane (dashed black boxes; bottom-left panels) were quantified (bottom-right panel). Images are representative of two independent experiments. Scale bar, 3 μm; zoom (inset) scale bar, 1 μm. WGA, wheat germ agglutinin. For (**a**), (**c**) and (**d**), F-actin was detected using phalloidin. Inverted lookup tables were applied. **e** Immunoprecipitation (IP) analysis of nesprin-2 protein complexes in Met4 cells. Anti-Mena- and anti-nesprin-2-reactive species detected by western blotting are indicated with arrowheads. High exposure of nesprin-2 blot of Met4 lysate (input) is also shown. Western blots are representative of three independent experiments. **f** Representation of nesprin-2 domains and the nesprin-2 constructs used for pulldown experiments. Constructs are based on the nesprin-2 giant isoform (nesprin-2G). **g**, **h** Pulldown analyses of protein complexes associated with exogenous GFP-tagged nesprin-2G(SR49–56) (**g**) and mini-nesprin-2GΔSR3–50 (**h**) in Met4 cells. Western blots are representative of three independent experiments. Ctrl control.

Mena11a was re-expressed because the main endogenous Mena-reactive band of Met4 cells migrates similarly with exogenous Mena11a (Supplementary Fig. 2a), suggesting that Met4 cells express the Mena11a variant. We verified this by sequence analysis of cDNA corresponding to the C-terminus of endogenous Mena obtained by reverse transcription of mRNA isolated from Met4 cells. We observed the presence of the additional 63 nucleotides that match the Mena11a variant (Supplementary Fig. 2b), thereby confirming the presence of endogenous Mena11a in Met4 cells, although expression of other Mena isoforms in Met4 cells is not excluded.

Co-immunoprecipitation of endogenous nesprin-2 revealed that depletion of Mena by CRISPR/Cas9-mediated knockout inhibited the interaction of nesprin-2 with actin (Fig. 3a) and suppressed the interaction between nesprin-2 and lamin A/C (Fig. 3b). These interactions were restored by re-expression of the Mena11a variant (Fig. 3a, b); the interaction between nesprin-2 and actin was increased by the overexpression of Mena11a, implying that the level of this biochemical complex may be dictated by the amount of Mena. Loss of Mena did not disrupt the interaction of nesprin-2 with the LINC complex component SUN2 (Supplementary Fig. 3a), suggesting that the associations of nuclear membrane proteins within the LINC complex are not regulated by Mena. These findings are consistent with Mena acting as a crucial facilitator/mediator of the interactions of nesprin-2 with actin and the nuclear lamina, but not with the rest of the LINC complex.

The interaction between the LINC complex and actin is reported to constrain nuclear shape in fibroblasts and carcinoma cells cultured in 2D[60] and 3D matrices[26]. We examined nuclear morphology by immunofluorescence and found that, in 3D collagen matrices, in which Mena also localised to the nucleus (Supplementary Fig. 3b), nuclear volume was significantly reduced in Mena-depleted cells (Fig. 3c, d). This implicates Mena in control of nuclear architecture in 3D environments via regulation of the nesprin-2–actin association. On 2D surfaces, depletion of nesprin-2 or disruption of perinuclear actomyosin is reported to increase nuclear height[61–63]. We observed an increase in nuclear height in Mena-depleted cells grown on 2D fibronectin matrices, which was rescued by re-expression of Mena11a (Fig. 3e, f). There was no significant difference in the relative proportion of F-actin to monomeric G-actin in cells depleted of Mena (Supplementary Fig. 3c, d), nor in total F-actin fluorescence intensity (Supplementary Fig. 3e, f), implying that total cellular actin polymerisation was not substantially perturbed by the loss of Mena in these cells. However, quantification of perinuclear F-actin directly above the nucleus revealed a decrease in nucleus-apical F-actin fluorescence intensity in cells depleted of Mena (Supplementary Fig. 3g, h), but no change in F-actin filament length or branching (Supplementary Fig. 3i), suggesting that the association of actin with the nucleus, but not nucleus-proximal actin filament organisation, is regulated by Mena. Together, these data are consistent with (i) Mena

being required for interaction between F-actin and the nucleus via the LINC complex component nesprin-2, and (ii) Mena controlling nuclear architecture via the F-actin–nesprin-2 connection in Met4 cSCC cells.

## Mena controls emerin tyrosine phosphorylation and linked gene expression

Given that LINC complex–actin cytoskeleton connections are known to mediate the transmission of tensile force generated by actomyosin contractility to the nucleus, we hypothesised that, by regulating these connections, Mena mediates nuclear mechanotransduction. Nuclear force transmission was recently shown to induce tyrosine phosphorylation of the nucleoskeleton- and LINC complex-binding protein emerin, thereby mediating an adaptive nuclear-stiffening mechanoresponse to tension in fibroblasts and HeLa cells[40]. To validate whether emerin tyrosine phosphorylation constitutes a readout for nuclear mechanotransduction in Met4 cells, we first disrupted actomyosin-based force contractility using the selective non-muscle myosin II inhibitor blebbistatin. Treatment with 10 μM or 50 μM blebbistatin for 2 h caused a dose-dependent reduction in emerin tyrosine phosphorylation (Supplementary Fig. 4a), consistent with previous studies[40]. Mena depletion also reduced emerin tyrosine phosphorylation in Met4 cells, and this was rescued by re-expression of Mena11a, with higher overexpression of Mena11a leading to higher levels of emerin tyrosine phosphorylation (Fig. 4a). It is known that keratinocytes react to tension by increasing histone H3 trimethylation at lysine-27 (H3K27me3)[64], and we found here that Mena depletion in Met4 cells resulted in a decrease in H3K27me3, which was rescued by re-expression of GFP-Mena11a (Supplementary Fig. 4b). Taken together, these data imply that Mena is required for relaying mechanical contractility cues to the nucleus by regulating the link between actin and the LINC complex component nesprin-2, and that this has consequences for transcription regulatory mechanisms, such as histone H3 trimethylation.

Upon mechanical stress, emerin is tyrosine-phosphorylated at both Y74 and Y95 (ref. [40]), which, together with Y59 phosphorylation, have been associated with a reduction in the binding of its LEM domain to the DNA-binding protein barrier-to-autointegration factor (BAF) in HeLa cells treated with pervanadate[65]. Emerin interaction with BAF mediates the recruitment of sections of chromatin known as lamina-associated domains (LADs) to the nuclear lamina, which has been associated with gene repression[66]. We therefore tested whether Mena-regulated changes in tyrosine phosphorylation of emerin could regulate the positioning of chromatin at the nuclear lamina. We performed multiplexed gene expression analysis of parental Met4 cells and their Mena-depleted counterparts (Met4 Mena$^{-/-}$), digitally profiling the expression of 770 genes associated with cancer progression (Supplementary Fig. 4c, d; Supplementary Data 3). We found that Mena depletion caused a statistically significant change in the

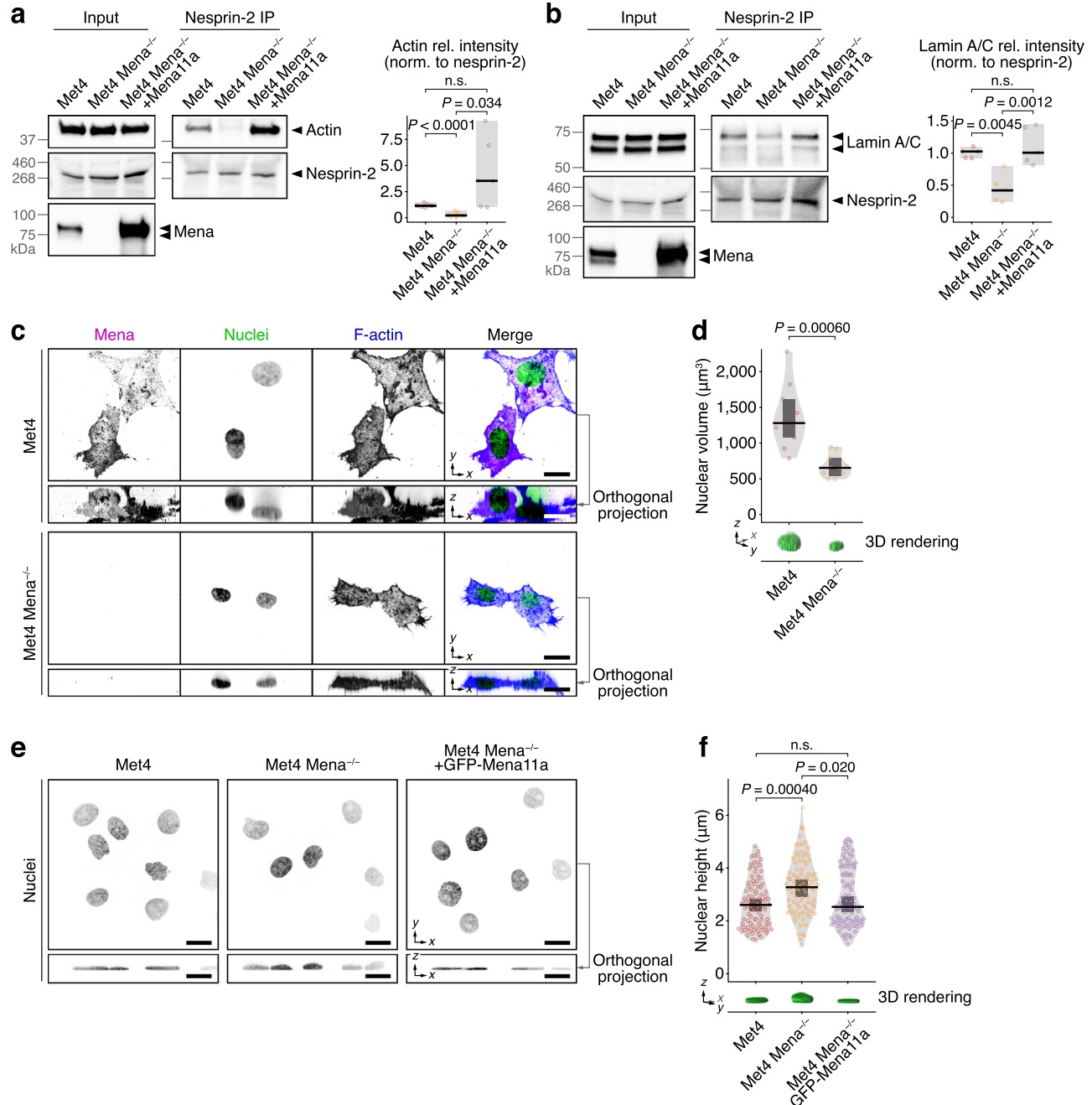

**Fig. 3 | Mena is required for nesprin-2 links to actin and lamin A/C and regulates nuclear morphology. a**, **b** IP analyses of nesprin-2 protein complexes in Met4 cells, Mena-depleted Met4 cells (Met4 Mena$^{-/-}$) and Met4 Mena$^{-/-}$ cells with Mena11a re-expressed (Met4 Mena$^{-/-}$ +Mena11a). Actin (**a**) and lamin A/C (**b**) were detected by western blotting; respective densitometric intensities were normalised (norm.) to nesprin-2 and expressed relative (rel.) to Met4 cells (right panels). Black bar, median; light grey box, range. Statistical analysis, Welch's one-way ANOVA with two-stage Benjamini–Krieger–Yekutieli correction ($n = 6$ independent experiments) for (**a**), one-way ANOVA with Tukey's correction ($n = 5$ independent experiments) for (**b**). **c** Spinning-disk confocal imaging of Met4 cells and Met4 Mena$^{-/-}$ cells in 3D collagen matrix. Orthogonal projections (*xz*) were extracted from 3D brightest-point projections. Nuclei were detected using DAPI; F-actin was detected using phalloidin. Inverted lookup tables were applied. Images are representative of two independent experiments. Scale bar, 20 µm. **d** Quantification of nuclear volume of cells in 3D matrix (see **c**). 3D volume renderings of exemplar nuclei detected using DAPI are displayed. Black bar, median; dark grey box, 95% confidence interval; light grey silhouette, probability density. Statistical analysis, two-sided Welch's *t*-test ($n = 10$ cells from $n = 2$ independent experiments). **e** Spinning-disk confocal imaging of Met4 cells, Met4 Mena$^{-/-}$ cells and Met4 Mena$^{-/-}$ +GFP-Mena11a cells on 2D fibronectin matrix. Orthogonal projections were extracted from 3D brightest-point projections. Nuclei were detected using DAPI. Inverted lookup tables were applied. Images are representative of four independent experiments. Scale bar, 20 µm. **f** Quantification of nuclear height of cells on 2D matrix (see **e**). 3D volume renderings of exemplar nuclei detected using DAPI are displayed. Black bar, median; dark grey box, 95% confidence interval; light grey silhouette, probability density. Statistical analysis, Kruskal–Wallis test with Dunn's correction ($n = 92$, 87 and 94 cells for Met4, Met4 Mena$^{-/-}$ and Met4 Mena$^{-/-}$ +GFP-Mena11a cells, respectively, from $n = 4$ independent experiments). n.s. not significant.

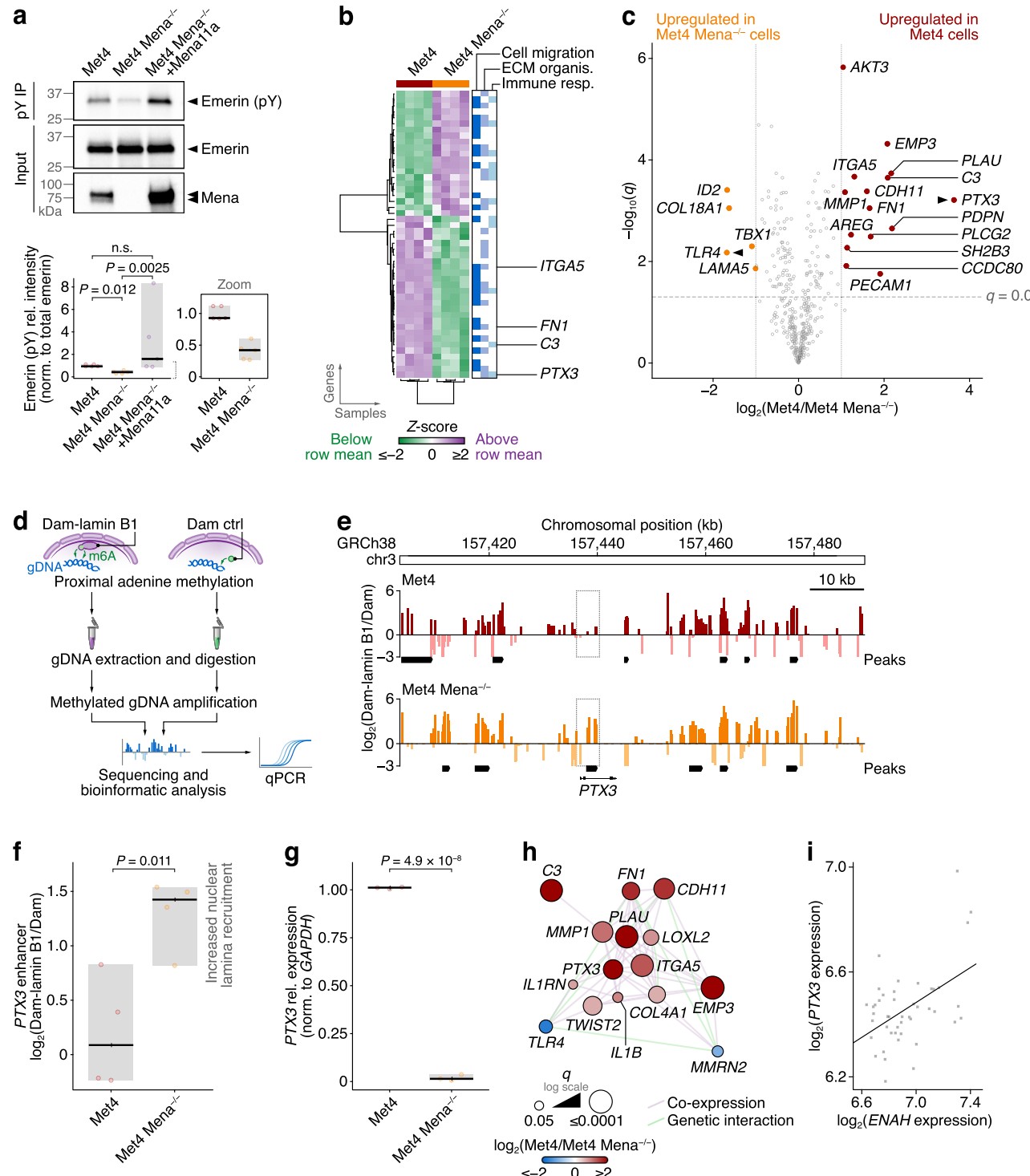

expression of 48 of the profiled cancer-associated genes (Fig. 4b, c; Supplementary Fig. 4e), which were broadly associated with regulation of cell adhesion and migration, ECM organisation and the immune response (Fig. 4b; Supplementary Fig. 4f; Supplementary Data 4). Of the dysregulated genes, the expression of 15 was significantly reduced by at least two-fold upon Mena depletion (Fig. 4c; Supplementary Data 4). Among these were the genes encoding fibronectin (*FN1*) and α5 integrin (*ITGA5*) (Fig. 4b, c; Supplementary Fig. 4f), consistent with a previously published study showing that *ENAH* expression correlates with *FN1* and *ITGA5* expression[67]. Mena loss also resulted in down-regulation in the expression of genes that encode immunomodulatory secretory proteins that are part of the complement system, such as

*PTX3* and *C3* (Fig. 4b, c; Supplementary Fig. 4f), suggesting that Mena may regulate immunosurveillance via the tumour-associated microenvironment.

We next addressed whether the Mena-dependent effects on some Mena-dependent transcriptional changes correlated with recruitment of LADs to the nuclear lamina that is associated with reduced tyrosine phosphorylation of emerin upon Mena depletion. We used DNA adenine methyltransferase identification (DamID) to map chromatin interactions with the nuclear lamina across the cSCC genome. Parental Met4 cells and their Mena-deficient counterparts were transduced with Dam or with Dam fused to nuclear lamina protein lamin B1 (Fig. 4d). Adenines in DNA regions in close proximity to lamin B1, and likely

**Fig. 4 | Mena loss suppresses emerin phosphorylation and regulates *PTX3* expression. a** Emerin tyrosine phosphorylation in Met4, Met4 Mena[−/−] and Met4 Mena[−/−] +Mena11a cells detected by phosphotyrosine (pY) IP and western blotting for emerin. Normalised densitometric intensities were expressed relative to Met4 cells (bottom panel). Black bar, median; light grey box, range. Inset (bottom-right panel) shows zoom of Met4 and Met4 Mena[−/−] cell quantification for clarity. Statistical analysis, Kruskal−Wallis test with two-stage Benjamini−Krieger−Yekutieli correction ($n = 5$ independent experiments). **b** Hierarchical cluster analysis of cancer progression genes significantly differentially regulated between Met4 and Met4 Mena[−/−] cells ($q < 0.05$, two-sided *t*-test with Benjamini−Hochberg correction; $n = 4$ independent biological replicates). Genes associated with cell migration, ECM organisation or the immune response are indicated (blue bars). **c** Volcano plot of cancer progression genes. Genes significantly differentially regulated between Met4 and Met4 Mena[−/−] cells by at least two-fold are labelled. Black arrowheads, the most differentially regulated gene in Met4 and Met4 Mena[−/−] cells, respectively. **d** Workflow for quantification of recruitment of specific regions of chromatin to the nuclear lamina by DamID. Green arrows, adenine methylation (m6A) in GATC motifs proximal to Dam. **e** Lamin B1 DamID sequencing (DamID-seq) tracks

generated from Met4 and Met4 Mena[−/−] cells. The highest-scoring putative enhancer region associated with *PTX3* (GeneHancer identifier GH03J157436) is indicated with a dotted box. Black bars, DamID-seq peaks (FDR < 5%; $n = 2$ independent experiments). Scale bar, 10 kb. **f** Nuclear lamina association of *PTX3* quantified by lamin B1 DamID-qPCR. Black bar, median; light grey box, range. Statistical analysis, two-sided Student's *t*-test ($n = 4$ independent experiments). **g** Quantification of *PTX3* expression by RT-qPCR. *GAPDH*-normalised gene expression was expressed relative to Met4 cells. Black bar, median; light grey box, range. Statistical analysis, two-sided Student's *t*-test ($n = 3$ independent experiments). **h** Subnetwork analysis of *PTX3*-associated genes from the gene expression analysis in (**c**). **i** Correlation of *ENAH* and *PTX3* expression in patient-derived cSCC cell lines (GEO series accession identifier GSE98767). Pearson correlation coefficient = 0.487; two-sided $P = 6.87 \times 10^{-4}$ ($n = 45$ samples from $n = 3$ independent replicates). Artwork in (**d**) was adapted from Byron, A., Bernhardt, S., Ouine, B. et al. Integrative analysis of multi-platform reverse-phase protein array data for the pharmacodynamic assessment of response to targeted therapies. Scientific Reports 10, 21985 (2020). https://doi.org/10.1038/s41598-020-77335-0.

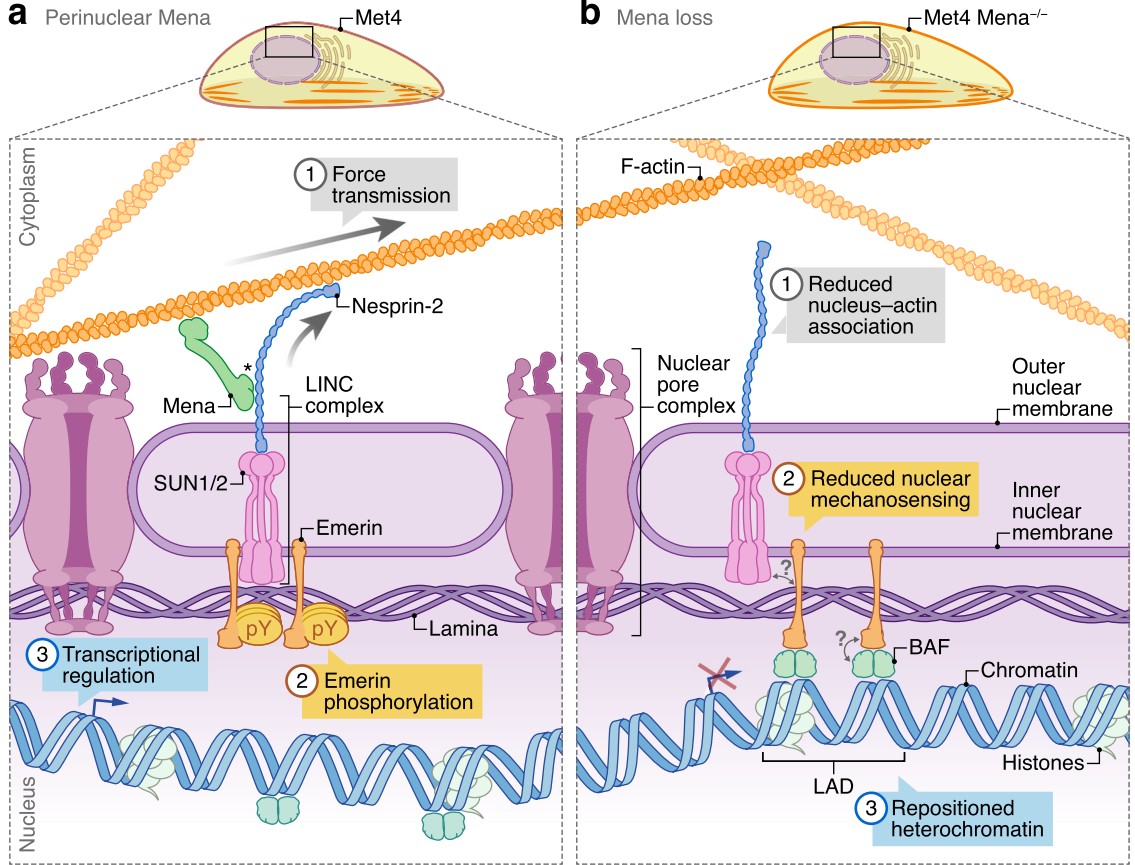

**Fig. 5 | Model of the proposed role for Mena at the nuclear membrane. a** Mena interacts with nesprin-2 at the outer nuclear membrane via the C-terminal SRs of nesprin-2. This potentiates the interactions of nesprin-2 with F-actin and lamin A/C, permitting force transmission from the actin cytoskeleton to the nuclear lamina (1). Actomyosin-based force maintains emerin tyrosine phosphorylation. Emerin tyrosine phosphorylation has been reported to prevent its LAP2−emerin−MAN1 (LEM) domain from binding to the DNA-binding protein barrier-to-autointegration factor (BAF), limiting chromatin repositioning to the nuclear lamina (2). This proposed mechanism allows genomic loci to favour chromatin decondensation and expression of specific genes, including, in metastatic Met4 cSCC cells, those involved in cancer progression, such as genes associated with cell adhesion and migration (e.g. *ITGA5*, *PLAU*), ECM organisation (e.g. *FN1*, *MMP1*) and the immune response (e.g. *PTX3*, *C3*) (3). **b** Mena loss

reduces the connectivity between F-actin and nesprin-2 (1), diminishing force transduction to the nuclear envelope via nesprin-2. Mena loss thus results in a reduction of emerin tyrosine phosphorylation (2), which has been reported to promote BAF binding to the emerin LEM domain, and enhances recruitment of heterochromatic lamina-associated domains (LADs) to the nuclear periphery (3). Chromatin repositioning is associated with transcriptional silencing of genes with regulatory elements in specific LADs (3), such as the immunomodulatory gene *PTX3* in cSCC cells. *, the nesprin-2-interacting region of Mena has not been determined. The hetero-oligomeric SUN−nesprin assembly is simplified for visualisation purposes. Question marks indicate unresolved potential mechanistic associations. Artwork was adapted from Byron, A., Griffith, B.G.C., Herrero, A. et al. Characterisation of a nucleo-adhesome. Nat Commun 13, 3053 (2022). https://doi.org/10.1038/s41467-022-30556-5.

proximal to emerin, were thus methylated (m6A) in GATC motifs by the tethered Dam as described previously[68]. The genomic fragments flanked by methylated GATC motifs were sequenced and the data converted to ratios of Dam-lamin B1 versus untethered Dam control normalised reads to produce genome-wide nuclear lamina association profiles (Fig. 4d). Upon Mena depletion, we observed a significant lamin B1 DamID peak (FDR 2.5%) at a region of chromosome 3 that corresponds to a predicted enhancer for *PTX3* (highest-scoring *cis*-regulatory element by GeneHancer analysis)[69], the most down-regulated gene upon Mena depletion, which exhibited increased association with the nuclear lamina in Mena-deficient cells compared to parental Met4 cells (Fig. 4e). Analysis of DamID fragments by qPCR confirmed an increase in nuclear lamina-associated chromatin spanning the *PTX3* enhancer locus in Mena-deficient cells (Fig. 4f), implying that the *PTX3* enhancer is recruited to the nuclear lamina upon Mena depletion. Predicted enhancers for other genes highly downregulated in Mena-deficient cells also exhibited increased association with the nuclear lamina in cells depleted of Mena (Supplementary Fig. 4g). In contrast, for genes upregulated in Mena-deficient cells, we observed loss of significant lamin B1 DamID peaks – which were present in parental Met4 cells (FDR < 5%) – at regions corresponding to their predicted enhancers (Supplementary Fig. 4g). Enhancers sequestered to the nuclear lamina can be associated with their inactivation[70]. Our findings are therefore consistent with the observation that *PTX3*, and other cell migration and immune response genes, are significantly dysregulated in Mena-depleted Met4 cells, as we showed by multi-plexed gene expression analysis (Fig. 4b, c; Supplementary Fig. 4f) and confirmed by RT-qPCR (Fig. 4g), via repositioning of particular LADs to the nuclear periphery. Moreover, previous studies have shown that *PTX3* expression is inducible by the interleukin IL-1β. The latter is downregulated in Mena-deficient cells, as determined by gene expression analysis (Supplementary Fig. 4f), and is part of a *PTX3* co-expression subnetwork along with several other Mena-regulated genes (Fig. 4h), suggesting another regulatory layer by which Mena could regulate *PTX3* expression. To confirm that Mena does commonly regulate the expression of *PTX3* in cSCC, we analysed publicly available microarray transcriptomics data from patient-derived cSCC samples (GEO series accession identifier GSE98767)[71] and found that *ENAH* and *PTX3* expression are marginally positively correlated (Pearson correlation coefficient = 0.487, $P = 6.87 \times 10^{-4}$) in human cSCC (Fig. 4i).

Taken together, our results imply that Mena regulates the expression of genes, at least in part, through its interaction with nesprin-2, which links the actomyosin contractile apparatus of the cell, via the LINC complex, to the nucleus (Fig. 5). This, in turn, elicits changes in emerin tyrosine phosphorylation at the nuclear lamina, and we identified a subset of genes, including *PTX3*, whose transcription is regulated by Mena as a result of altered chromatin organisation (Fig. 5).

## Discussion

Mena is an actin regulatory adhesion protein, mostly defined at actin-rich structures, such as lamellipodia, filopodia, focal adhesions and cell–cell contacts, as well as stress fibres, along which it forms periodic puncta[72]. Here, we identified a role for Mena at the nuclear membrane, where it regulates actin–nuclear lamina associations, nuclear architecture, chromatin repositioning and gene expression. Our results show that an adhesion protein can modulate transcription via direct signalling across the nuclear membrane.

We found, using a network biology approach in patient-derived cSCC cells, that Mena was part of a functional adhesome module consisting of F-actin-binding and regulatory proteins. Subnetwork association of Mena and the LINC complex component nesprin-2 led us to uncover a function of Mena at the perinucleus, whereby it interacts with the C-terminal SRs of nesprin-2 and potentiates F-actin binding to nesprin-2 (Fig. 5). This, in line with other observations[26,38,56],

supports the view that actin-binding proteins may mediate the interaction of nesprin SRs with actin[73], and we show here that Mena is one of these. We also found that Mena regulates nuclear morphology in cSCC cells, consistent with its function of mediating the interaction of actin with nesprin-2. It is possible that Mena–nesprin-2 interactions may induce conformational changes in nesprin-2 that modulate the ability of nesprin-2 to bind actin, but we did not assess this in the present study. By modulating interactions of nesprin-2, Mena controls phosphorylation of the nuclear membrane protein emerin, repositioning of chromatin at the nuclear periphery and the regulation of some genes involved in tumour progression (proposed model in Fig. 5). How Mena is transported to the perinuclear region to elicit these functions, and the nature of common and distinct pools of Mena, is not known.

Delineating a role for Mena at the perinucleus, we show here that Mena is required for the mechanoresponse of emerin (Fig. 5). Emerin is a member of the LEM domain protein family, forming an extensive interaction network with nuclear lamina and nucleoplasmic proteins. We found that Mena depletion results in the reduction of emerin tyrosine phosphorylation and an increased association of a *PTX3* enhancer with the nuclear lamina. Interestingly, emerin phosphorylation can also lead to remodelling of its interaction network at the nuclear lamina[74]. For instance, the interaction of emerin with histone deacetylase 3 (HDAC3) results in increased HDAC3 deacetylase activity, which is associated with gene repression[75]. Emerin phosphorylation is postulated to affect its interaction with HDAC3 at the nuclear lamina, thereby modulating mechanoresponsive gene expression[74], and we found that Mena loss results in a reduction of H3K27me3, a histone modification associated with gene repression. A previous study has shown that mechanical strain induces H3K27me3 and reduces active RNA polymerase II occupancy in keratinocytes, and this is dependent on a perinuclear emerin–non-muscle myosin II–F-actin axis[64]. Further investigating the role of Mena in this emerin–actomyosin axis may provide insights into the mechanism of Mena-dependent H3K27me3 modification.

We found that Mena status controls gene expression, including genes that influence cell migration and ECM organisation, processes that shape the tissue microenvironment and are dysregulated in tumorigenesis and metastasis. A subset of Mena-dependent genes, including *PTX3*, were also linked to immune modulation, another central process in tumour growth. We found that Mena regulates the association of a *PTX3* enhancer with the nuclear lamina and the expression of the gene encoding IL-1β, an inducer of *PTX3* expression. Tumour-promoting and tumour-suppressive roles for PTX3 have been described, while elevated levels of PTX3 correlate with either poor prognosis or grade of malignancy in several cancers[76]. Though the role of PTX3 in the context of cSCC is unknown, regulation of the expression of *PTX3*, and other immune response genes, by perinuclear activities of Mena supports observations that focal adhesion proteins have nuclear or nucleus-associated functions in regulating immuno-modulatory proteins[31]. Indeed, we have demonstrated previously that nuclear FAK induces the expression of proinflammatory cytokines and chemokines, including IL-33 and CCL5, which inhibit antitumour immunity and drive tumour growth in vivo[77,78].

The systems-level changes to the adhesome of cSCC cells we report here, including the enrichment of Mena in the adhesome of metastatic Met4 cSCC cells when compared to their non-metastatic counterparts from the same patient, suggest putative molecular mechanisms through which adhesion protein networks may drive cancer progression. Using a network analysis approach, we found that Mena clusters within an actin-binding/regulatory module that is also upregulated in metastatic cSCC cells compared to primary cSCC cells, and our analyses connected Mena to the LINC complex component nesprin-2. Although we have not functionally implicated the Mena–LINC complex association in the metastatic process in this study, Mena, and its role in regulating cancer cell motility, have been

linked with tumour progression, invasion and metastasis[79–82]. How the subcellular localisation of Mena, and other adhesion proteins, influences malignant progression requires further investigation.

In conclusion, this work establishes a connection between the adhesome component Mena and the LINC complex, through which Mena regulates actin–nuclear lamina interactions, nuclear architecture and chromatin organisation at the nuclear periphery, fine-tuning cell migration and immune gene expression, as demonstrated by the immunomodulatory gene *PTX3*, in malignant keratinocytes.

## Methods

### Cell culture
Human cSCC Met1 (subclone 1) and Met4 (subclone 1) cells[83,84] were cultured in RM+ medium consisting of high-glucose Dulbecco's modified Eagle's medium (DMEM; Sigma-Aldrich, #D6546) and Ham's F-12 nutrient mixture (Life Technologies, #21765-037) (3:1 DMEM:Ham's F-12 ratio) supplemented with 10% (v/v) foetal bovine serum (FBS; Life Technologies, #10270-106), 0.4 µg/ml hydrocortisone (STEMCELL Technologies, #7904), 3 mM L-glutamine (#200049), 5 µg/ml transferrin (#T2252), $1 \times 10^{-10}$ M cholera toxin (#C8052), 10 ng/ml epidermal growth factor (#E2644), 5 µg/ml insulin (#I9278) and $2 \times 10^{-11}$ M liothyronine (all Sigma-Aldrich, #T6397). HEK293T cells (a gift from Noor Gammoh, University of Edinburgh, Edinburgh, UK) were cultured in high-glucose DMEM supplemented with 10% (v/v) foetal bovine serum. Cells were grown at 37 °C in humidified 5% (v/v) $CO_2$. Cells routinely tested negative for mycoplasma.

For 3D culture in collagen, Met1 or Met4 cells were trypsinised and collected by centrifugation at $200 \times g$ for 5 min. Cells ($5 \times 10^4$) were resuspended in 2–3 mg/ml rat-tail collagen I in RM+ medium and seeded on 23-mm FluoroDish cell culture dishes (World Precision Instruments). The collagen with embedded cells was allowed to set at 37 °C until collagen contraction was observed (~30 min). Another aliquot of collagen-embedded cells was added to the contracted matrix and allowed to set at 37 °C. Collagen-embedded cells were then incubated in RM+ medium.

### Cell line generation and transfection
Met1 cells were derived from primary cSCC and Met4 cells from distant metastatic cSCC from the same immunosuppressed patient[43,83]. Ethical approval (REC reference 08/S1401/69, 5 November 2008) for this study was obtained from the East London and City Health Authority local ethics committee, and the study was conducted according to the Declaration of Helsinki Principles. Met1 subclone 1 and Met4 subclone 1 were established following cell culture and used for the experiments herein (referred to herein as Met1 cells and Met4 cells, respectively).

Met4 Mena$^{-/-}$ cells were generated by CRISPR/Cas9 technology. Guide oligonucleotides targeting exon 2 of *ENAH* (Supplementary Table 1) containing an *Afl*II restriction site and protospacer adjacent motif were designed and synthesised (Integrated DNA Technologies). The guide RNA (gRNA) cloning vector (Addgene, #41824) was linearised by *Afl*II digestion, and guide oligonucleotides were annealed to the linearised gRNA cloning vector by Gibson Assembly. Constructs were transformed into chemically competent DH5α cells and selected on agar plates containing kanamycin. Positive gRNA guides were identified by sequencing. Met4 cells were co-transfected with *ENAH* gRNA and hCas9 (Addgene, #41824) using the Amaxa human keratinocyte Nucleofector kit (Lonza, #VPD-1002) according to manufacturer's instructions, with a Nucleofector (Lonza) set to programme T-024. Three rounds of nucleofection were performed, allowing cells to recover and grow to 60% confluency between nucleofection rounds.

Cells were transiently transfected with GFP-nesprin-2G(SR49–56) (which contains the eight C-terminal nesprin-2 SRs 49–56), GFP-mini-nesprin-2GΔSR3–50 (also known as GFP-mini-nesprin-2G SR51–56, which contains SRs 51–56 of nesprin-2; based on the nesprin-2 giant isoform, nesprin-2G)[26], pEGFP-C1 or pcDNA-Mena11a (a gift from

Francesca Di Modugno, Regina Elena National Cancer Institute, Rome, Italy). All GFP tags used were EGFP. Cells were transiently transfected with constructs using the X-tremeGENE HP DNA transfection reagent (Roche) according to manufacturer's instructions. Briefly, vectors were prepared (2:1 transfection reagent:DNA ratio) and incubated for at least 30 min prior to transfection of $2 \times 10^6$ cells seeded on fibronectin-coated cell culture dishes. Cells were used for experiments 60 h after transfection.

GFP-Mena11a was expressed in Met4 Mena$^{-/-}$ cells using pMSCV-EGFP-Mena11a (a gift from Frank Gertler, Massachusetts Institute of Technology, Cambridge, MA, USA; ref. [80]). Phoenix amphoteric cells were transfected with pMSCV-EGFP-Mena11a using Lipofectamine 2000 (Life Technologies) according to manufacturer's instructions. Eighteen hours after transfection, viral production was induced with fresh medium supplemented with 20% (v/v) FBS. Virus was harvested 48 and 72 h after transfection, and the viral supernatant was filtered through a 0.45-µm Millex-HA filter (Millipore). Filtered viral supernatant was supplemented with 5 µg/ml polybrene (Millipore) and added to Met4 Mena$^{-/-}$ cells for 24 h. Two rounds of transduction were performed. GFP-Mena11a-expressing cells were selected by fluorescence-activated cell sorting.

### Blebbistatin treatment
Cells were treated with 10 µM and 50 µM blebbistatin (in DMSO; Calbiochem) or DMSO (vehicle control) for 2 h at 37 °C in humidified 5% (v/v) $CO_2$.

### Immunoprecipitation and pulldown
For nesprin-2 immunoprecipitation (IP), cells were washed twice with ice-cold PBS and lysed in NETN lysis buffer (100 mM NaCl, 20 mM Tris-HCl, pH 7.5, 0.5 mM EDTA, 0.5% (v/v) NP-40) supplemented with protease inhibitors. Cells were homogenised by 15 strokes through a 23 G syringe needle. Homogenate was clarified by centrifugation at $16,000 \times g$ for 10 min at 4 °C. Supernatant (1 mg protein) was incubated overnight with rotation at 4 °C with either nesprin-2 antibody (Abcam, #ab217057) or anti-rabbit IgG (Cell Signaling Technology, #2729) and Protein G Dynabeads (Invitrogen, #10003D). Beads were washed three times with lysis buffer, and isolated immune complexes were eluted with Laemmli sample buffer for 10 min at 95 °C.

For phosphotyrosine IP, immune complexes were isolated as above, except cells were lysed in RIPA buffer (50 mM Tris-HCl, pH 7.5, 150 mM NaCl, 1% (v/v) Triton X-100, 0.5% (w/v) sodium deoxycholate) supplemented with protease and phosphatase inhibitors. Supernatant was incubated with anti-phosphotyrosine antibody (clone PY20; BD Transduction, #610000) or, for blebbistatin treatment experiments, anti-phosphotyrosine antibody coupled to M-270 Epoxy Dynabeads (Invitrogen, #14301) according to manufacturer's instructions.

For GFP-Trap pulldown, cells were treated with 3 mM dimethyl 3,3′-dithiobispropioimidate (DTBP; Thermo Fisher Scientific, #20665) for 5 min at 37 °C, which was then quenched with 200 mM Tris-HCl (pH 7.8) for 10 min at room temperature. Cells were lysed in RIPA buffer as above, except cell lysates were sonicated after homogenisation. Supernatant was incubated with GFP-Trap magnetic agarose (ChromoTek) according to manufacturer's instructions. Beads were washed three times with RIPA buffer, and isolated complexes were eluted with Laemmli sample buffer for 10 min at 95 °C.

### Western blotting
Protein concentration was measured by Pierce BCA protein assay (Thermo Fisher Scientific). Protein was supplemented with Laemmli sample buffer to a final concentration of 0.5–1.0 mg/ml and boiled for 10 min at 95 °C. Proteins were resolved by SDS–polyacrylamide gel electrophoresis using 4–15% Mini-PROTEAN TGX gels (Bio-Rad) or, to resolve anti-nesprin-2-reactive bands, 3–8% Tris-acetate gels (Life Technologies).

Proteins were transferred to nitrocellulose membrane by semi-dry transfer (Trans-Blot Turbo transfer system; Bio-Rad) or, to transfer nesprin-2, by wet transfer in Tris-glycine buffer (Bio-Rad) supplemented with 15% (v/v) methanol. Membranes were blocked with 5% (w/v) milk or 5% (w/v) bovine serum albumin (BSA) in Tris-buffered saline (TBS). Membranes were probed with the following primary antibodies (all Cell Signaling Technology, diluted 1:1000, unless otherwise stated): anti-αV integrin (clone EPR16800; Abcam, #ab179475), anti-actin (clone 13E5; #4970; or clone 8H10D10; #3700), anti-cytochrome c oxidase subunit 4 (clone 4D11-B3-E8; #11967), anti-emerin (clone D3B9G; #30853), anti-FAK (#3285), anti-FAK pY397 (#3283), anti-GAPDH (clone D16H11; #5174), anti-GFP (BioVision, #3999-100), anti-H3K27me3 (clone C36B11; #9733), anti-histone H3 (clone D1H2; #4499), anti-lamin A/C (#2032), anti-Mena (Atlas, #HPA028448), anti-myosin light chain 2 (clone D18E2; #8505), anti-myosin light chain 2 pS19 (#3675), anti-nesprin-2 (Abcam, #ab217057). Membranes were incubated with anti-rabbit or anti-mouse horseradish peroxidase-conjugated secondary antibodies (Cell Signaling Technology, #7074, diluted 1:10,000, or #7076, diluted 1:5000, respectively) and were visualised using a ChemiDoc MP Imaging System (Bio-Rad) and analysed using Image Lab (version 5.2.1) (Bio-Rad).

### Immunofluorescence staining

For cells on 2D substrate, cells were seeded on fibronectin-coated coverslips or glass-bottomed chamber slides (Ibidi, #80827) and allowed to spread overnight. Cells were fixed with 4% (w/v) paraformaldehyde for 15 min, washed twice with PBS and incubated in 0.1 M glycine for 5 min. For STED, cells were fixed with 4% (w/v) paraformaldehyde for 15 min, washed three times with Hank's balanced salt solution (HBSS), incubated with 5 µg/ml biotinylated wheat germ agglutinin (Sigma-Aldrich) in HBSS for 10 min, washed twice with HBSS, fixed with 4% (w/v) paraformaldehyde in HBSS for 15 min and washed twice with HBSS. For all techniques, cells were then permeabilised with 0.2% (v/v) Triton X-100 in PBS for 5 min, washed twice with PBS and blocked with 2% (w/v) BSA in PBS for 1 h. Cells were incubated with anti-Mena (clone A351F7D9; Merck Millipore, #MAB2635) or anti-nesprin-2 (Abcam, #ab204308), diluted 1:200 in 2% (w/v) BSA in PBS, overnight at 4 °C. Cells were washed six times with PBS and incubated with anti-rabbit secondary antibodies conjugated to Alexa Fluor 594 (Thermo Fisher Scientific, #A-11072) or Abberior STAR RED (Abberior, #STRED-1002-500UG; STED), anti-mouse secondary antibodies conjugated to ATTO 647 N (Rockland Immunochemicals, #610-156-121) or Abberior STAR 580 (Abberior, #ST580-1001-500UG; STED), phalloidin conjugated to Alexa Fluor 488 (Thermo Fisher Scientific, #A12379) or Biotium CF680R (Cambridge Bioscience, #BT00048; STED) and streptavidin conjugated to Abberior STAR 488 (Abberior, #ST488-0120-1MG; STED), diluted 1:400 (1:800 for phalloidin) in 0.05% (v/v) Tween 20 in PBS, for 45 min in the dark. Cells were washed six times with PBS, incubated with 4% (w/v) paraformaldehyde for 5 min, washed with PBS and incubated with 0.1 M glycine for 10 min. For SoRa spinning-disk confocal, cells were incubated with phalloidin conjugated to ATTO 647 N (Sigma-Aldrich, #65906), diluted 1:400 in 0.05% (v/v) Tween 20 in PBS, for 1 h in the dark and then washed three times with 0.05% (v/v) Tween 20 in PBS. Coverslips were mounted with anti-fade Fluoroshield containing DAPI (Sigma-Aldrich) or ProLong glass anti-fade mountant (Thermo Fisher Scientific; STED).

For cells in 3D collagen matrices, matrix-embedded cells were fixed with formaldehyde-PIPES buffer (4% (w/v) formaldehyde, 100 mM PIPES, pH 6.8, 10 mM EGTA, pH 8.0, 1 mM MgCl$_2$, 0.2% (v/v) Triton X-100) for 30 min and permeabilised with 0.2% (v/v) Triton X-100 in TBS (TBS-TX) for 30 min. Autofluorescence of the collagen

matrix was quenched with two 10-min washes with 0.5 mg/ml NaBH$_4$. Cells were blocked with 2% (w/v) BSA in TBS-TX for 4 h. Cells were incubated with anti-Mena (Atlas, #HPA028448), diluted 1:200 in 2% (w/v) BSA in TBS-TX, overnight at 4 °C. Cells were washed six times with TBS-TX and incubated with anti-rabbit secondary antibody conjugated to Alexa Fluor 594 and phalloidin conjugated to ATTO 647 N, diluted 1:400 in TBS-TX. Cells were washed six times with TBS-TX, and coverslips mounted on collagen matrices with anti-fade Fluoroshield containing DAPI.

### Immunofluorescence imaging

Spinning-disk confocal images were acquired on a Dragonfly multi-modal imaging platform (Andor Technology) with 405-, 488-, 561- and 637-nm excitation laser lines, and 450/50, 525/50, 620/60 and 698/70 filters, using a 100× oil-immersion objective. Data were collected in spinning-disk mode, with a 25-µm pinhole, on a iXon 888 EMCCD camera (Andor Technology) using 1×1 binning and 4× frame averaging. Confocal images were acquired with z-step size of 0.5 µm using a piezo positioning system (Mad City Labs). For comparative experiments, images were collected with the same settings for all cell lines.

Optical pixel reassignment imaging was performed on a CSU-W1 SoRa super-resolution spinning-disk confocal scanning unit (Nikon Instruments) with 405- and 638-nm excitation laser lines, and 447/50 and 708/75 filters, using a 100× 1.49NA oil-immersion objective (Nikon Instruments) and refractive index-matched immersion oil (Nikon Instruments). Data were collected in Yokogawa spinning-disk mode, with a 50-µm pinhole, on a Prime 95B Scientific CMOS camera (Photometrics) using 1×1 binning and no frame averaging. Image z-step size was set to 0.100 µm. Image deconvolution was performed using NIS-Elements Advanced Research software (version 5.21.03) (Nikon Instruments).

3D-SIM images were acquired on an N-SIM super-resolution microscope (Nikon Instruments) with 488-, 561- and 644-nm excitation laser lines, and 520/35, 593/40 and 655/40 filters, using a 100× 1.49NA oil-immersion objective (Nikon Instruments) and refractive index-matched immersion oil (Nikon Instruments). Cells were imaged using a DU-897X-5254 camera (Andor Technology). Image z-step size was set to 0.120 µm as recommended by the manufacturer's software. For each focal plane, 15 images (5 phases, 3 angles) were captured. 3D-SIM image processing, reconstruction and analysis were performed using the N-SIM module in NIS-Elements Advanced Research software (version 4.6).

Gated-STED imaging was performed on a TCS SP8 STED 3X super-resolution confocal microscope (Leica Microsystems) using an HC PL APO 100× 1.40NA STED WHITE oil-immersion objective (Leica Microsystems). Samples were excited using a supercontinuum white light laser and detected using a HyD hybrid detector (Leica Microsystems), with excitation and detection wavelengths set accordingly (Supplementary Table 2). STED depletion was performed using 592 nm and 775 nm depletion lasers (Leica Microsystems) (Supplementary Table 2). Images were acquired in xzy scan mode, with a pixel size of 13 nm and 20× frame averaging.

### Image analysis

For nuclear morphology analyses, regions of interest (nuclei) were generated on the basis of DAPI staining. Nuclear height was measured from the minimum length of the object-orientated bounding box that fully enclosed the nucleus, implemented in Imaris (version 9.2.0) (Oxford Instruments). Nuclear volume was measured and rendered in Fiji (version 1.52p)[85]. DAPI-stained images were background subtracted using a sliding paraboloid set to 50 pixels and thresholded using the Moments method. Voxel measurements were limited to the thresholded region.

For quantification of STED-derived xz images, the respective intensity of each channel, $I$, where $\{I \in \mathbb{R} : I_{min} \leq I \leq I_{max}\}$, was

transformed to $I_N$, where $\{I_N \in \mathbb{R} : 0 \leq I_N \leq 1500\}$, using the following linear normalisation:

$$I_N = (I - I_{min})\frac{1500}{I_{max} - I_{min}} \qquad (1)$$

For co-occurrence analyses, Manders' coefficients were calculated using Imaris (version 9.8.0). Automatic thresholding was used to determine the thresholds of overlapping voxels derived from different channels. For cytoplasmic co-occurrence, nesprin-2 voxels representing the nuclear membrane were segmented, and the voxels within the segmented surface were set to 0 for all channels. For nuclear co-occurrence, only nesprin-2 voxels representing the nuclear membrane were segmented and compared to overlapping voxels derived from a different channel.

For nucleus-apical F-actin analyses, z-slices of the phalloidin staining at the apical region of the cell were stacked to produce a maximum-intensity z-projection of nucleus-apical actin filaments. The regions of interest (apical nuclear surfaces) were generated on the basis of DAPI staining. A bandpass filter was applied to the region of interest, followed by an unsharp mask (default mode) using Fiji (version 1.53c). Actin filaments were thresholded and gaps and holes were filled using the Close function. Intensity was measured by redirecting the thresholded image to the corresponding unprocessed region of interest. For branching and length analyses, the thresholded actin filaments were skeletonised, and the resulting skeleton was analysed using the Skeleton and AnalyzeSkeleton functions.

### Actin fractionation

F-/G-actin ratio was determined using a G-actin/F-actin in vivo assay biochem kit (Cytoskeleton, Inc.). Briefly, $2 \times 10^6$ cSCC cells were seeded on 10-cm cell culture dishes and incubated overnight. Cells were washed with PBS at 37 °C and lysed in 1 ml lysis and F-actin stabilisation buffer containing 1 mM ATP and protease inhibitor cocktail. Cell lysates were homogenised with 15 strokes through a 25-gauge needle, incubated for 10 min at 37 °C and clarified by centrifugation at $350 \times g$ at room temperature. The clarified homogenate was centrifuged at $100,000 \times g$ for 1 h at 37 °C, after which the supernatant containing G-actin was recovered and the pellet containing F-actin was solubilised with F-actin depolymerisation buffer. The fractions were analysed by western blotting; actin was probed with a monoclonal anti-actin antibody (Cell Signaling Technology, #3700).

### Adhesome isolation

For IAC isolation[9,10], $3 \times 10^6$ cSCC cells were seeded on 15-cm cell culture dishes coated with 10 µg/ml human fibronectin (Corning, #356008) and incubated overnight. Cells were crosslinked with 3 mM DTBP in DMEM/Ham's F-12 nutrient mixture (1:1 DMEM:Ham's F-12 ratio) for 5 min at 37 °C, which was then quenched with 200 mM Tris-HCl (pH 6.8) for 5 min at room temperature. Cells were washed with 200 mM Tris-HCl (pH 6.8) and lysed with ice-cold IAC extraction buffer (0.05% (w/v) NH₄OH, 0.5% (w/v) Triton X-100 in PBS). Cell bodies were removed by hydrodynamic pressure applied with a Waterpik device (Waterpik Ultra Water Flosser WP-120), and IAC proteins were isolated using adhesion recovery buffer (125 mM Tris-HCl, pH 6.8, 1% (w/v) SDS, 150 mM dithiothreitol). Isolated IACs were analysed by western blotting (see above) or processed for LC-MS/MS analysis (see below).

For proteomic analysis, isolated IACs were precipitated with four volumes of acetone (−20 °C), incubated at −80 °C overnight and collected by centrifugation at $16,000 \times g$ for 20 min at 4 °C. Protein pellets were washed with acetone (−20 °C), collected by centrifugation and air-dried. Protein pellets were resuspended in 0.2% (w/v) RapiGest (Waters) for 2 h, incubated at 90 °C for 10 min and subjected to in-solution tryptic digestion at 37 °C overnight. Peptides were acidified with trifluoroacetic acid (~1% (v/v) final concentration), desalted on homemade C18 StageTips and resuspended in 0.1% (v/v) tri-fluoroacetic acid. Purified peptides were analysed by LC-MS/MS (see below).

### MS data acquisition

Peptides were analysed by LC-MS/MS using an UltiMate 3000 RSLCnano system coupled online to a Q Exactive Plus Hybrid Quadrupole-Orbitrap mass spectrometer (both Thermo Fisher Scientific). Peptides were injected onto a C18-packed emitter in buffer A (2% (v/v) acetonitrile, 0.5% (v/v) acetic acid) and eluted with a linear 120-min gradient of 2–45% (v/v) buffer B (80% (v/v) acetonitrile, 0.5% (v/v) acetic acid). Eluting peptides were ionised in positive ion mode before data-dependent analysis. The target value for full scan MS spectra was $3 \times 10^6$ charges in the 300–1650 $m/z$ range, with a resolution of 70,000. Ions were fragmented with normalised collision energy of 26, selecting the top 12 ions. A dynamic exclusion window of 30 s was enabled to avoid repeated sequencing of identical peptides. The target value for MS/MS spectra was $5 \times 10^4$ ions in the 200–2000 $m/z$ range, with a resolution of 17,500. All spectra were acquired with 1 microscan and without lockmass. Two technical replicate injections were performed per sample for each of three independent biological experiments.

### MS data analysis

Label-free quantitative analysis of MS data was performed using MaxQuant (version 1.5.3.17)[86]. Peptide lists were searched against the human UniProtKB database (version 2015_09) and a common contaminants database using the Andromeda search engine[87]. Cysteine carbamidomethylation was set as a fixed modification; methionine oxidation and protein N-terminal acetylation were set as variable modifications (up to five modifications per peptide). MS data from two technical replicate analyses were merged for each corresponding independent biological experiment in the peptide search. Peptide and protein FDRs were set to 1%, determined by applying a target-decoy search strategy using MaxQuant. Proteins with only shared peptides, which could not be unambiguously identified by unique peptides, were assigned to a protein group, and the first protein accession of a protein group sorted by MaxQuant was extracted for further analysis (Supplementary Data 1). At least two peptides, including at least one unique peptide or razor peptide (a shared peptide assigned to the protein group with the largest number of peptide identifications), were required for protein identification. Enzyme specificity was set as C-terminal to arginine and lysine, except when followed by proline, and a maximum of two missed cleavages were permitted in the database search. Minimum peptide length was seven amino acids, and at least one peptide ratio was required for label-free quantification. Proteins matching to the common contaminants database (excluding potentially relevant UniProtKB accessions P02533, P02538, P04264, P05787, P06396, P08729, P08779, P13645, P13647, P19012, P23142, P35443, P35527, P35908, P49747, P78385, P99999, Q04695, Q14CN4, Q2M2I5, Q5D862, Q86YZ3, Q9H4B7) or the reverse database and matches only identified by site were omitted.

Label-free quantification intensities for proteins quantified in at least two out of three biological replicate analyses of either Met1 or Met4 IACs were binary-logarithm-transformed. Data were normalised globally by robust linear regression using Normalyzer (version 1.1.1)[88]. Values missing in all biological replicates of an experimental group were imputed using a first-pass single-value imputation, whereby the local minimum logarithm-transformed intensity value across a given replicate was used to impute missing-not-at-random missing values[89]. Remaining missing values were imputed using a second-pass predictive mean matching imputation using the MICE R package (version 3.9.0)[90]. Statistical comparisons between experimental groups were carried out using two-sided Student's t-tests (for experimental groups with equal variance; F-test) or two-sided Welch's t-tests (where experimental groups displayed unequal variance; F-test) with Benjamini–Hochberg

correction. Proteins enriched in an experimental group by at least two-fold with $q < 0.05$ were considered significantly differentially enriched.

## Functional enrichment analyses

Proteins were classified as actin-cytoskeletal proteins if they were annotated with Gene Ontology terms GO:0015629, GO:0030864, GO:0005884 or GO:0001725; actin-binding proteins were those annotated with terms GO:0003779 or GO:0051015; proteins involved in actin organisation were those annotated with terms GO:0030036, GO:0032956, GO:0030866, GO:0007015, GO:0110053, GO:0051639, GO:0043149 or GO:0051492; proteins involved in cell migration were those annotated with terms GO:0016477, GO:0030334, GO:0030335, GO:0001755, GO:0050900, GO:0043534, GO:0043536, GO:0043542, GO:0051451, GO:0010634, GO:0002042, GO:1901164 or GO:0048870; proteins involved in ECM organisation, composition or binding were those annotated with terms GO:0030198, GO:0022617, GO:0090091, GO:1901201, GO:0050840, GO:0031012, GO:0005201, GO:0062023 or GO:1904466; proteins involved in the immune response were those annotated with terms GO:0006955, GO:0045087, GO:0002250, GO:0002683 or GO:0002768.

Over-representation analyses were performed using WebGestalt (version 2019)[91]. To reduce redundancy of enriched functional categories, where stated, gene sets were clustered according to Jaccard index and classified with representative terms using affinity propagation via the APCluster R package implemented in WebGestalt[91,92]. Significant enrichment of categories compared to the curated meta-adhesome reference set (the intersection set of the meta-adhesome and the literature-curated adhesome) was determined by hypergeometric test with Benjamini–Hochberg correction ($q < 0.05$), and categories with enrichment ratio ≥1 were displayed. Size of dataset overlap with respective gene sets was mapped to data point fill saturation.

## Interaction network analysis

For analysis of the cSCC adhesome, a protein interaction network was constructed as an undirected graph using curated physical protein-protein interactions from the BioGRID database and integrated with the quantitative MS data[93]. To identify the most relevant functional active module, the maximal-scoring subgraph was calculated and extracted from the graph using a diffusion-flow emulation model implemented in the BioNet R package (version 1.46.0)[50]. P-values derived from statistical comparisons between experimental groups (two-sided Student's or Welch's t-tests; see above) were converted into FDR-corrected scores based on signal–noise decomposition. Scores were assigned to corresponding nodes (proteins) to generate a weighted graph (FDR 5%)[93]. The problem of finding a maximum-weight connected subgraph was transformed into a prize-collecting Steiner tree (PCST) problem, and the exact solution to the latter was computed using integer linear programming implemented in the Heinz R package (version 1.63)[94].

For clustering the active module using the Louvain modularity maximisation method, edges were weighted by Spearman rank correlation coefficient +1 (ref. [93]). We optimised the Girvan–Newman modularity quality function using the igraph R package (version 1.2.5) [https://igraph.org]. For clustering the active module using the constant Potts model, edges were weighted by Spearman rank correlation coefficient[93]. The problem of clustering was reformulated as a problem of finding the ground state of the Potts spin-glass model with a tuneable resolution parameter, γ (ref. [95]). We scanned γ to maximise the Surprise quality function using the louvain Python package (version 0.6.0) [https://pypi.org/project/louvain]. Active modules were visualised using Cytoscape (version 3.6.1)[96] via the CyREST API[97]. Molecular functions enriched in protein clusters were determined by Gene Ontology over-representation analysis ($q < 0.05$, hypergeometric test with Benjamini–Hochberg correction), and

clusters were annotated with the two most significant functional categories; for protein clusters with fewer than two significant functional categories, clusters were annotated manually with a single representative term. Kinless and connector hubs were represented in network graphs as large nodes.

For analysis of the core cSCC adhesion proteins or gene expression data, composite functional association networks were constructed using GeneMANIA (version 3.5.1; human interactions)[98] in Cytoscape (version 3.8.0)[96]. For gene expression data, edges were weighted according to evidence of co-expression or genetic interaction. Networks were clustered using the edge-weighted force-directed algorithm in the Prefuse toolkit[99].

## Unsupervised learning

Binary, agglomerative hierarchical cluster analysis of Z-transformed protein or mRNA abundance was performed using Cluster 3.0 (C Clustering Library, version 1.54)[100]. Euclidean distance matrices were computed using average linkage, and clustering results were visualised using Java TreeView (version 1.1.5r2)[101]. For sample correlation analysis, Pearson correlation coefficient-based distance matrices were computed using complete linkage. Dimensionality reduction using principal component analysis was performed using R, and plots were annotated with 95% confidence ellipses.

## RNA isolation

RNA was extracted from Met4 and Met4 Mena$^{-/-}$ cells using an RNeasy Mini kit (Qiagen) following the manufacturer's instructions. Homogenisation was performed by passing cells (15×) through a 23-gauge syringe. RNA was eluted from silica membranes in 30 µl RNase-free water. An additional elution step was performed using the eluate of the first elution step to increase RNA yield. RNA sample concentrations were measured using a NanoDrop 2000c spectrophotometer (Thermo Fisher Scientific). All RNA samples had a ratio of absorbance at 260 nm and 280 nm (A260/280) of ~2.0.

## *ENAH* sequencing

For sequencing of *ENAH* from Met4 cells, 1 µg of total RNA was used to synthesise cDNA using a SuperScript First-Strand kit (Invitrogen). cDNA (Supplementary Table 1) was amplified using Phusion High-Fidelity DNA Polymerase (New England Biolabs) in PCR reactions consisting of an initial denaturation step of 30 s at 95 °C, followed by 30 cycles at a denaturation temperature of 95 °C (30 s per cycle), an annealing temperature of 60 °C (30 s per cycle) and an extension temperature of 72 °C (1 min per cycle). A final extension step was performed at 72 °C (10 min). PCR product was analysed on a 1% (w/v) agarose gel, excised and purified using a gel extraction kit (Qiagen). PCR product was cloned into pCR-Blunt II-TOPO (Invitrogen) by blunt-end ligation according to manufacturer's instructions and transformed into DH5α cells. Plasmid was isolated using a QIAprep Miniprep kit (Qiagen) and sequenced using M13 sequencing primers (Supplementary Table 1).

## Multiplexed gene expression analysis

Total RNA was analysed using the 770-gene human nCounter Pan-Cancer Progression panel (NanoString Technologies) according to manufacturer's instructions. nCounter data were normalised to synthetic positive control targets and to housekeeping genes using nSolver Analysis software (version 4.0) (NanoString Technologies). A minimum mean count threshold was set to 20 counts, yielding a unimodal distribution of binary-logarithm-transformed counts. To remove between-batch variation, we used the pseudoreplicates approach of the removing unwanted variation-III normalisation method[102], applying $k = 2$. Statistical comparisons between experimental groups were carried out using two-sided Student's t-tests (for experimental groups with equal variance; F-test) or two-sided Welch's

*t*-tests (where experimental groups displayed unequal variance; *F*-test) with Benjamini−Hochberg correction.

## Lamin B1 DamID

For lentiviral transduction of Met4 and Met4 Mena$^{-/-}$ cells for lamin B1 DamID[103], HEK293T cells were co-transfected with psPAX2 (a lentiviral packaging plasmid), pMD2.G (which encodes vesicular stomatitis virus G protein) (both gifts from Val Brunton, University of Edinburgh) and a vector encoding either Dam fused to lamin B1 (pLgw Dam-lamin B1) or untethered Dam (pLgw Dam) (both gifts from Eric Schirmer, University of Edinburgh). Transfection medium was changed for fresh growth medium 18 h after transfection. Viral supernatant was harvested 72 h after transfection, filtered through a 0.45-µm Millex-HA filter (Millipore), diluted in DMEM (2:1 DMEM:supernatant ratio), supplemented with polybrene and added to Met4 and Met4 Mena$^{-/-}$ cSCC cells seeded on 6-well cell culture plates. cSCC cells were transferred to 15-cm dishes (1 × 10$^6$ cells per dish) and incubated for 60 h, followed by genomic DNA (gDNA) extraction using a DNeasy Blood & Tissue Kit (Qiagen) according to manufacturer's instructions.

For methylated GATC library preparation, gDNA (300 ng) was digested using *Dpn*I (New England Biolabs) for 6 h at 37 °C, followed by *Dpn*I inactivation for 20 min at 80 °C. dsAdR adaptor ligation was performed using T4 DNA ligase (New England Biolabs) overnight at 16 °C, followed by ligase inactivation for 10 min at 65 °C. Digestion using *Dpn*II (New England Biolabs) was then performed for 1 h at 37 °C, followed by *Dpn*II inactivation for 20 min at 65 °C. The reaction mixture was purified using a PCR purification kit (Qiagen) and eluted in RNase-free water. Initial PCR amplification using AdR_PCR primers (Supplementary Table 1) (ref. [103]) and SYBR Select master mix was performed with a first cycle of extension at 72 °C for 10 min, followed by one cycle of denaturing at 94 °C (1 min), annealing at 65 °C (5 min) and extension at 72 °C (15 min) and then four cycles of denaturing at 94 °C (1 min), annealing at 65 °C (1 min) and extension at 72 °C (10 min). The optimum number of remaining cycles was determined by plotting linear Rn (passive reference dye-normalised reporter signal) versus cycle number and calculating the number of cycles corresponding to one-third of the height of the sigmoid curve to avoid PCR duplication. Methylated GATC gDNA fragments were purified using solid-phase reversible immobilisation beads and eluted in RNase-free water. GATC gDNA fragments (50 ng) were treated with Nextera Tn5 transposase (Illumina) for 5 min at 37 °C with shaking. The transposase reaction was stopped by adding binding buffer from the PCR purification kit, and tagged DNA fragments were eluted in RNase-free water. DNA fragments were indexed and barcoded (Ad2.*n*) and PCR-amplified using Ad1_noMX and Ad2.*n* primers (Supplementary Table 1) (ref. [104]) and SYBR Select master mix, with initial holding temperature of 72 °C (5 min) followed by 98 °C (30 s). The optimum number of cycles was determined by qPCR (see below) at a denaturing temperature of 98 °C (10 s), annealing temperature of 63 °C (30 s) and extension temperature of 72 °C (1 min). Library fragments were purified using solid-phase reversible immobilisation beads and eluted in RNase-free water. Next-generation sequencing was performed using the NextSeq 500/550 high-output kit (version 2.5; 150 cycles) (Illumina, #20024907) on the NextSeq 550 platform (Illumina, #SY-415-1002).

## Sequencing data analysis

Nextera adaptor trimming was performed using TrimGalore (version 0.6.3) [https://www.bioinformatics.babraham.ac.uk/projects/trim_galore] with the minimum required overlap with the adaptor sequence set to 5. Single-end read binning against the human reference genome (GRCh38) was performed using BBSplit, part of the

BBMap package (version 38.11) [https://sourceforge.net/projects/bbmap]. Binned reads were aligned (single-end mapping) to the human reference genome using Bowtie 2 (version 2.3.4.3)[105]. Mapped reads were sorted and indexed using SAMtools (version 1.6). The human reference genome was binned into GATC motifs fragments[106] and expanded by 250 bp on either side (500-bp fragment window). Single-end reads were imported as a GenomicRanges object in R, and the average number of reads within the 500-bp GATC window tilling the entire human chromosomes was computed. Read values were normalised to the total number of mapped reads (excluding decoy chromosomes). Ratios of Dam-lamin B1 versus untethered Dam control normalised reads were calculated and binary-logarithm-transformed, and data were visualised using the Integrated Genome Browser (version 9.1.4)[107]. Peak FDR analysis was performed using the find_peaks script [https://github.com/owenjm/find_peaks], with peak intensity threshold set to a minimum quantile of 75%, stepping of 0.0005 and 15 iterations (peak FDR < 5%).

## qPCR analysis

RT-qPCR (1 µl of cDNA) was performed on a StepOnePlus Real-Time PCR system using SYBR Select master mix (both Thermo Fisher Scientific). Reactions consisted of 10 min at 95 °C, followed by 40 cycles of 30 s at 95 °C, 30 s at 60 °C and 30 s at 72 °C. *PTX3* gene expression (Supplementary Table 1) was normalised to *GAPDH* (Supplementary Table 1), calculating ΔΔCt values.

For DamID-qPCR, methylated GATC gDNA fragments (2.5 µl) were subjected to qPCR using SYBR Select master mix with qPCR reaction conditions as detailed above. Primers (Supplementary Table 1) were used to amplify the putative *PTX3* enhancer at genomic region 157,437,977–157,438,893 bp of chromosome 3 (human reference genome GRCh38). Statistical significance was assessed by two-sided Student's *t*-test.

## Microarray data analysis

The expression of *ENAH* and *PTX3* was assessed across a panel of patient-derived cSCC samples (GEO series accession identifier GSE98767)[71]. Data were binary-logarithm-transformed and normalised by cyclic loess using the limma R package (version 3.11)[108]. Scatter plot line-of-best-fit was generated using R, and linear correlation was assessed by Pearson correlation coefficient.

## Statistics and reproducibility of experiments

Distributions of residuals were tested for normality using the Shapiro−Wilk test. Statistical significance of data with normally distributed residuals was calculated using a two-sided Student's or Welch's *t*-test (for comparing two unmatched groups) or an ordinary one-way ANOVA with Tukey's correction or Welch's one-way ANOVA with two-stage Benjamini−Krieger−Yekutieli correction (for comparing three unmatched groups). Statistical significance of data with distributions of residuals that departed from normality was calculated using a Kruskal−Wallis test with Dunn's correction or two-stage Benjamini−Krieger−Yekutieli correction (for comparing three unmatched groups). For proteomic data analysis, proteins quantified in at least two out of three independent biological replicates for at least one experimental condition were further analysed, and significantly differentially abundant proteins were determined using two-sided Student's *t*-tests (for experimental groups with equal variance; *F*-test) or two-sided Welch's *t*-tests (for experimental groups with unequal variance; *F*-test) with Benjamini−Hochberg correction. Significantly differentially transcribed genes were determined using the same statistical tests as used for proteomic data analysis. For functional enrichment analyses, significantly enriched terms were determined using a hypergeometric test with Benjamini−Hochberg correction. No statistical method was used to predetermine

sample size. Data plots were generated using Cytoscape, Excel (Microsoft), Integrated Genome Browser, Java TreeView, PlotsOfData[109] or R.

## Reporting summary

Further information on research design is available in the Nature Portfolio Reporting Summary linked to this article.

## Data availability

The MS-based proteomics data generated in this study have been deposited in ProteomeXchange via the PRIDE partner repository[110] with dataset accession identifier PXD021492. The DamID sequencing data generated in this study have been deposited in the Gene Expression Omnibus[111] with GEO series accession identifier GSE159598. The protein sequence data used in this study are available in the Uni-ProtKB database with release identifier 2015_09. The human reference genome data used in this study are available from the Genome Reference Consortium with release identifier GRCh38. All other data supporting the findings of this study are available within the paper and its Supplementary Information and Source Data files. Requests for materials should be addressed to Adam Byron. Source data are provided with this paper.

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

## Acknowledgements

We are grateful to Val Brunton, Francesca Di Modugno, Noor Gammoh, Frank Gertler and Eric Schirmer for reagents. We thank the University of Edinburgh Institute of Genetics and Cancer (IGC) Mass Spectrometry Facility for LC-MS/MS data acquisition; Alison Munro and the IGC Host and Tumour Profiling Unit Microarray Services for multiplexed gene expression analysis; Lizzie Freyer and the IGC Flow Cytometry Facility for flow cytometry analysis; Stephen Brown, Jeffrey Joseph and the IGC DNA Sequencing Facility for DNA sequencing; Amy Davies, Laura Murphy, Matt Pearson and the IGC Advanced Imaging Resource and Edinburgh Super Resolution Imaging Consortium for assistance with microscopy and image analysis; Richard Clark and the Edinburgh Clinical Research Facility for next-generation sequencing; Ainara Cabodevilla for assistance with lentiviral transduction; Laura Gómez-Cuadrado for assistance with RT-qPCR; Noor Gammoh for discussions and assistance with CRISPR/Cas9; Colin McLean for discussions and assistance with network analysis; Katerina Petelova for discussions and exploratory data analysis; Val Brunton, Gareth Inman, Roza Masalmeh, Bryan Serrels and Andy Sims for discussions. The work was funded by Cancer Research UK (grants C157/A15703 and C157/A24837 to M.C.F.). Development of the cSCC cell lines was funded by Cancer Research UK (grant A13044) and the European Research Council (grant 250170). W.A.B. was funded by a Medical Research Council University Unit grant (grant MC_UU_00007/2). M.P. was funded by the Medical Research Council (grant MR/R008264/1). J.D.A. was funded by a European Union Horizon 2020 Framework Programme for Research and Innovation (grant 945539; Human Brain Project SGA3). Y.K. was funded by a Wellcome Trust Discovery Award (grant 217120/Z/19/Z to W.A.B.). The STED system

at the Edinburgh Super Resolution Imaging Consortium was supported by the Wellcome Trust (grant 208345/Z/17/Z). The work made use of the resources provided by the Edinburgh Compute and Data Facility (ECDF), University of Edinburgh [https://www.ecdf.ed.ac.uk]. For the purpose of open access, the authors have applied a CC-BY public copyright licence to any author accepted manuscript version arising from this submission.

## Author contributions

M.C.F. and A.B. conceived and co-ordinated the project; F.L.M.C., B.G.C.G., A.P.W., J.D.A., M.P., I.M.L., C.M.P., A.v.K., W.A.B., M.C.F. and A.B. designed the experiments and interpreted the results; F.L.M.C., B.B., B.G.C.G., A.E.P.L., Y.K., J.C.W., M.L. and J.V. performed the experiments; F.L.M.C., B.B. and A.B. analysed the data and prepared the figures; F.L.M.C., M.C.F. and A.B. wrote the paper; all authors commented on the manuscript and approved the final version.

## Competing interests

The authors declare no competing interests.

## Additional information

[1]Cancer Research UK Scotland Centre, Institute of Genetics and Cancer, University of Edinburgh, Edinburgh EH4 2XR, UK. [2]MRC Human Genetics Unit, Institute of Genetics and Cancer, University of Edinburgh, Edinburgh EH4 2XU, UK. [3]Edinburgh Super Resolution Imaging Consortium, Institute of Biological Chemistry, Biophysics and Bioengineering, School of Engineering and Physical Sciences, Heriot-Watt University, Edinburgh EH14 4AS, UK. [4]Advanced Imaging Resource, Institute of Genetics and Cancer, University of Edinburgh, Edinburgh EH4 2XU, UK. [5]Simons Initiative for the Developing Brain, School of Informatics, University of Edinburgh, Edinburgh EH8 9LE, UK. [6]Randall Centre for Cell and Molecular Biophysics, King's College London, London SE1 1UL, UK. [7]Division of Molecular and Clinical Medicine, School of Medicine, University of Dundee, Dundee DD1 9SY, UK. [8]Institute of Dentistry, Barts and the London School of Medicine and Dentistry, Queen Mary University of London, London E1 2AT, UK. [9]Division of Molecular and Cellular Function, School of Biological Sciences, Faculty of Biology, Medicine and Health, University of Manchester, Manchester Academic Health Science Centre, Manchester M13 9PT, UK. [10]Present address: Department of Oncology, Medical Sciences Division, University of Oxford, Oxford OX3 7DQ, UK. ✉e-mail: adam.byron@manchester.ac.uk

