## [Peer Review File · Nature Communications]

Mena regulates nesprin-2 to control actin–nuclear lamina associations, trans-nuclear membrane signalling and gene expressionREVIEWER COMMENTS

Reviewer #1 (Remarks to the Author):

The Authors investigate the role of the protein called Mena in trans-nuclear membrane signaling. They find that Mena is enriched in IACs in cSCC cells and claim that it is connected to the LINC complex component Nesprin-2 with which it interacts and co-localizes at the nuclear envelope of metastatic cells. Mena seems to potentiate the interactions of Nesprin-2 with the actin cytoskeleton and the nuclear lamina and its depletion causes altered nuclear morphology, reduced tyrosine phosphorylation of the nuclear-membrane protein Emerin and downregulated expression of the immunomodulatory gene PTX3 via recruitment of its enhancer to the nuclear periphery. As such, the manuscript reveals novel connection between Mena and the LINC complex, it shows that Mena might regulate actin-nuclear lamina interactions and in this way affecting nuclear architecture and gene expression of cancer-relevant genes (especially those involved in cancer progression). In this way, it shows a direct link between proteins involved in adhesion and gene expression regulation, which might be disrupted or perturbed in cancer.

The study is very well presented and touches upon multiple very relevant topics such as metastasis, signal transduction and nuclear/chromatin organization. However, in its current form, the manuscript sends out too many messages which are not substantiated and provides a rather confusing picture, which would need to be framed into a more coherent one. Below I provide comments/suggestions which, if addressed, should strengthen the claims in the manuscripts.

Major comments:

- The manuscript is very well written and clearly presents the data with a logical flow when it comes to individual figures. However, the big picture gets lost on the way and towards the end the reader can get confused by what the message really is. The authors introduce the manuscript by discussing the role of adhesion complexes in cancer progression and metastasis, yet this aspect is lost throughout the text aside from the fact that experiments are performed on Met4 cell line, which is derived from metastatic tumor. I would suggest either explaining the findings in the context of metastasis and how they specifically affect its onset or else stay clear from it altogether. On this note, it would be interesting to provide a more in-depth comparison with how Mena works in Met1 cell line, which is derived from the primary tumor (this is lacking completely).
- If the emphasis on metastasis is to remain in the manuscript I would suggest informing the reader about its mutation / expression status in various primary vs metastatic tumors.

- The entire manuscript is based on the putative interaction between Mena and nesprin-2 and once this is established early on in the manuscript all the phenotypes the Authors observe upon Mena depletion are then explained by the loss of this interaction. The fact that Mena localizes throughout the cell and likely has other functions than the ones proposed by the Authors is ignored. It might very well be that the effects the Authors see (which are convincing on their own) stem from the effects of Mena depletion on the plasma membrane or/and the cytoskeleton, and I could easily imagine that if the cytoskeleton is perturbed that will affect nuclear morphology. Therefore, the Authors should try to disentangle the effects Mena's interaction with nesprin-2 might have on the nuclear morphology (and related aspects), from the effects Mena might have when not in complex with nesprin-2. For this reason, the comparison of Mena depletion in Met1 and Met4 cells would be particularly useful. Related to this, would the depletion of nesprin-2 mimic the effects exerted by the depletion of Mena on the nucleus?

- The Authors should comment on the big picture of the changes they observe in actin-linked proteins (Fig 1f) or ECM or cell migration proteins (Fig. 4d) given that proteins belonging to all these groups get upregulated in both Met1 in comparison to Met4, Met4 in comparison to Met1, and in Mena wt with respect to Mena-/- (and vice versa). I understand that the Authors chose one protein to focus on but since the manuscript addresses the question of metastasis and the role of the actin – lamina association in this process, seeing these results might be confusing without any guidance.

- The Authors have performed a lot of screens, have undertaken multiple high throughput approaches, yet from each approach they typically chose to focus on one selected candidate protein or a gene (as in the case of the DamID data and the PTX3 gene). Is PTX3 the only upregulated gene whose upregulation can be explained by the DamID data?

- [Fig 2a-b] The authors highlight the localization of Mena at the focal adhesion and at the nuclear membrane and claim co-localization between Nesprin-2 and Mena. This is not obvious from the image presented. Rather, the two signals are abutted. The Authors should be more accurate here. Given that the entire manuscript is built on the interaction between Mena and nesprin-2 I would recommend strengthening these claims perhaps by performing an even higher resolution imaging. What is most confusing though is that in the image shown it is not clear whether the proximity between Mena and actin occurs at the plasma membrane or at the nuclear membrane. Could the Authors indicate clearly where the plasma membrane would be in this image? Also, is the depletion of the Mena signal at the basal side of the nucleus/cell shown on this picture representative of all the cells? Could the Authors comment on it?

- The model the Authors propose in Figure 5 points to massive and multiple effects caused by Mena removal: 1) detachment of the actin network entirely from the nuclear envelop, 2) lack of Nesprin-2 – actin interaction, 3) lack of Nesprin-2 – Sun1/2 interaction, 4) lack of Sun1/2 with emerin interaction, 5) lack of emerin phosphorylation, 6) changes in the chromatin marks and 7) chromatin repositioning close to the nuclear membrane. To my understanding the Authors did not provide proofs for points 1, 3, 4 and

commented only on one enhancer changing its proximity to the membrane (with regards to point 7). Hence, the model in my view is far-stretched. If there is a complete loss of connectivity between the actin cytoskeleton and the nuclear envelope I would expect even more drastic changes to the morphology of the cell and nucleus. In order to prove such drastic changes at the nuclear envelope EM images are recommended.

- Could the Authors provide information on how long do the cells survive after the depletion of Mena?
- The claims about the function of Mena at the nuclear membrane would be stronger if the Authors could show a direct interaction of Mena with nesprin-2 or other proteins of the complex. Also the microscopy study of the co-localization between Mena and Nesprin-2 would benefit from approaches such as PLA.

Minor Points:

- [line 163 and Fig 1 legend] The Authors use the following expression on a few occasions: “Actin-associated proteins significantly differentially regulated between Met1 and Met4 IACs”. I believe that what they refer to are simply differences in abundance, which do not say per se anything about differences in the regulation. I would recommend to use a more accurate description here.
- [line 346] The Pearson coefficient of 0.487 is not necessarily indicative of correlation. The Authors should add a word of caution, stating that the correlation they identified is very weak (borderline).
- [line 378] I believe the authors meant “PTX3 enhancer” and not “PTX3”.
- [Fig 2b] No quantification of the Mena and Nesprin-2 co-localization is provided. Is the juxtaposition of the signal a result of Mena being present all over the cells? What is the enrichment score at the perinuclear region? How does this compare to the Met1 cells?
- [Fig 2c and Fig. 2e, f] Could the Authors comment on why in the nesprin-2 IP the two bands of Mena seem to be equally strong in the case of wt nesprin-2 (Fig 2c), while in the case of the truncations one of them seems much stronger (Fig 2e, f)?
- What is its expression status of PTX3 in Met1 cells (as well as its enhancer localization).
- [Fig 3a-b] Given the high variability of the Met4 Mena-/-+ Mena11a signals for both Actin and Lamin A/C it would be useful to also measure at the same time Mena in these pulled down complexes. Is there a reason why the Authors did not probe for Mena in these IP experiments? It would be interesting to see if the variability in the Actin or Lamin A/C signals were dependent on the Mena signal in the corresponding complexes.
- I suggest moving Fig 4a to Supplementary Figures.

Reviewer #2 (Remarks to the Author):

In their manuscript, Chee et al. investigated the systems-level changes in the integrin adhesome during the metastatic progression of a patient-derived cutaneous squamous cell carcinoma (cSCC). Their investigation revealed that the actin-regulator mammalian enabled (mena) is enriched in integrin adhesion complexes (IACs) in metastatic cSCC cells. By computationally constructing a protein interaction network of the cSCC adhesome using curated physical protein-protein interactions from the BioGRID database, the authors found that mena was proximal in an actin-binding/regulation sub-network to the actin-binding outer nuclear membrane KASH protein nesprin-2. The authors went on to show that mena colocalizes with nesprin-2 at the nuclear envelope and the two proteins form a biochemical complex as demonstrated by co-immunoprecipitation and pull-down experiments. Chee et al. then provide evidence to suggest that mena is required for the ability of nesprin-2 to interact with the actin cytoskeleton as well as the nuclear lamina. In addition, mena knock-out (KO) cSCCs displayed altered nuclear morphology and nuclear mechanotransduction as well as down-regulated expression of the immunomodulatory PTX3 gene. Moreover, the recruitment of the PTX3 enhancer to the nuclear periphery was elevated in mena KO cSCCs relative to control cells. In summary, this work suggests that mena performs several unexpected, yet related, functions at the cytoplasm surface of the nuclear envelope where it appears to work together with nesprin-2-containing LINC complexes to control the physical coupling of the nucleus and the actin cytoskeleton, nuclear morphology, chromatin organization, and cancer gene expression. While the implications of this work are quite novel and exciting, I cannot recommend that it be accepted for publication in its current form due to the following major and minor issues:

Major Issues:

1) The title states, “mena regulates the LINC complex”. However, there is little evidence presented in this work to support this claim. As shown in Figure 5, LINC complexes are nuclear envelope-spanning molecular bridges composed of the outer and inner nuclear membrane KASH and SUN proteins, respectively. Yet, the authors neglect to address the potential impact of mena on either of the widely expressed mammalian SUN proteins: SUN1 and SUN2. For example, the authors should determine if mena-depletion impacts the ability of nesprin-2 to interact with SUN1 and/or SUN2 as well as the nuclear envelope localization of nesprin-2, SUN1, and SUN2.

2) While the model provided by the authors in Figure 5A suggests that mena directly interacts with nesprin-2G at the outer nuclear envelope. Figure 5A also suggests that the authors know the part of mena that interacts with nesprin-2G. However, they do not present any data to support either suggestion. The authors really should determine if mena interacts directly with nesprin-2. They should then map the mean-binding site on nesprin-2 and vice versa. Armed with this information, the authors could then directly test the physiological relevance of the mena-nesprin-2 interaction by determining if a mena construct harboring a mutation that prevents its ability to interact with nesprin-2 can rescue the

nuclear morphology and emerin tyrosin phosphorylation phenotypes observed in the mena KO cSCCs. This information is really important for two reasons: 1) The sub-network analysis presented in Figure 1H shows that mena likely interacts with nesprin-2 through hPrp4; and 2) nesprin-2 was not identified in the meta-adhesome reported by Horton et al. (2015), while nesprin-1, nesprin-3, SUN1, and SUN2 were.

3) Ena/VASP proteins promote actin filament assembly at several subcellular locations by acting as processive actin polymerases. Yet, the authors of this work seem to be proposing that mena controls the coupling of the nucleus with the actin cytoskeleton solely by promoting the ability of nesprin-2G to interact with perinuclear actin filaments. If this were true, I would anticipate that the re-expression of a mena construct harboring a mutation that disables its polymerase activity without impacting its ability to interact with actin in the mena-KO Met4 cells would rescue their nuclear morphology and emerin tyrosin phosphorylation phenotypes. In addition, it would be good for the authors to quantitatively determine if the absence of mena has an impact on the organization of perinuclear actin in the Met4 cells.

4) In Figure 3A and 3B, the authors show the results of co-immunoprecipitation experiments suggesting that the interaction between nesprin-2 and actin or A-type lamins is disrupted in the absence of mena. While these results appear to be compelling, I would strongly caution the authors against drawing too many conclusions from them without support from orthogonal experimental evidence (i.e. FRET, FRAP, or co-localization). The presence of actin or A-type lamins in a co-immunoprecipitation should be concerning due to their ability to form mesh-like polymers that may non-specifically trap proteins resulting in them being co-immunoprecipitated. It should also be said that the effect of mena-depletion on the nesprin-2-A-type lamin interaction could either be due to mena impacting the ability of nesprin-2 to directly interact with SUN1/2 or A-type lamins. Given these results, are the authors suggesting that the interaction of nesprin-2G with mena induces an allosteric conformational change that modulates its ability to interact with either actin or A-type lamins? If so, they need to be more explicit.

5) In Figure 2a the nuclear envelope localization of mena in Met4 cells grown in 2D is clear, while this localization is apparently lost when Met4 cells are grown in 3D as shown in Figure 3C. Moreover, it appears that the nuclear envelope localization of mena in 2D cultured Met4 cells correlates with the levels of nesprin-2 present within the nuclear envelope. Based on these conflicting results, it is unclear to me how mena is being recruited to the nuclear envelope in the sCC cells in a nesprin-2-dependent manner. Therefore, does mena lose its ability to localize to the nuclear envelope in 3D cultured cells because nesprin-2 no longer is found at the nuclear envelope in cells grown under these conditions? It would also be good to know if disrupting the nuclear envelope localization of nesprin-2 by either knocking-down nesprin-2 or over-expressing a LINC complex-disrupting dominant negative construct (e.g. the KASH domain of nesprin-2) displaces mena from the nuclear envelope. This information would be critical for the authors to be able to distinguish between mena working through nesprin-2G-containing LINC complexes or focal adhesions.

6) In Figure 3C, the authors show that the volume of nuclei in mena-KO Met4 cells is significantly reduced relative to control cells. In Figure 3D, the authors show that nuclear height is significantly increased in mena-KO cells relative to Met4 cells expressing endogenous mena or mena-KO cells expressing GFP-mena11a. However, the F-actin images shown in Figure 3C suggest that the cytoplasmic area and volume of mena-KO Met4 cells may be reduced relative to control cells. Since nuclear and cytoplasmic volumes scale linearly in cells, the authors need to also quantify the effect of mena-KO on cytoplasmic volume. This is particularly important given the established role for mena as a regulator of actin polymerization, which is critical for cellular spreading and nuclear height control.

7) While the dam-ID results presented in Figure 4 are compelling, the authors need to provide support from orthogonal experimental evidence for the repositioning of the PTX3 enhancer within the nucleoplasm (e.g. FISH or dCas9-mediated imaging of genomic loci). Moreover, the authors should show if re-expression of GFP-mena11a in their mena-KO cells rescues the phenotypes reported in Figures 4G-H. Since they have the GFP-mena11a-expressing mena-KO cells (see Figs. 3A-B), this request should not be too difficult. Moreover, it would be good to know if the expression levels of PTX3 are also decreased in cells lacking nesprin-2 or expressing a LINC complex-inhibiting dominant negative construct.

8) Why do the authors only report the effect of deleting mena from Met4 cells on the positioning of PTX3's enhancer relative to lamin B1? Do the enhancers of the other genes that are up-regulated in Met4 cells behave similarly to the PTX3 enhancer? Conversely, what happens to the enhancers of the genes that are up-regulated in the mena-KO Met4 cells?

Minor Issues:

1) Human gene names are written in all caps and italics.

2) Page 4, Line 93: The authors state that nesprin-1, -2, -3, and -4 make up "the outer nuclear membrane nesprin family proteins". However, the authors should also include KASH5 and LRMP in this list.

3) The drawings of the GFP-nesprin-2G SR49-56 and Δ SR3-50 constructs shown in Figure 2D seem to have large stretches of light purple that are not present in the drawing of full-length nesprin-2G shown above. What are these light purple stretches?

4) The labels of Figures 2E and 2F should reflect the names of the GFP-tagged nesprin-2G constructs used to perform the pull-down experiments.

5) The name of the GFP-nesprin-2GΔSR3-50 should be changed to GFP-mini-nesprin-2GΔSR3-50, which is its original name per Jayo et al., 2016 Dev Cell where it was described.

6) I do not believe that the statement “mena localizes immediately adjacent to nesprin-2” (page 4, line 114) can be made without quantification or with the ~300 nm axial resolution afforded by Nikon SIM under optimal conditions.

7) Please identify in the Materials and Methods section the fluorescence filters used in the authors' Dragonfly and Nikon SIM imaging systems.

Reviewer #3 (Remarks to the Author):

The authors compare two BSCC variants from the same patient; one from the initial tumor (Met1) and the second from a metastatic tumor in a lymph node (Met4). Of the many proteins that are different between the two lines, they focus on Mena which is upregulated in the metastatic Met4 line. Mena interacts with Nesprin 2 and localizes to the nuclear membrane. There are changes in the expression profiles, nuclear shape, emerin phosphorylation and H3 methylation with the loss of Mena. The exact mechanisms of these changes are not detailed but a lot of circumstantial evidence is provided. Although I don't feel comfortable in evaluating the nodes in their modeling, there is clearly an important role for Mena. Missing are cell migration assays and direct measurements of metastasis; however, this is a lot to expect in an already data heavy paper. The major message of this paper is that Mena interacts with Nesprin 2 and their interaction is implicated in the nuclear morphology and expression pattern changes with Mena depletion. This is an important set of findings even though the cell biological implications of the changes have not been fully explored. Because of the large amount of data that may be useful to other labs, I recommend publication with only minor revisions.

Results

Gene clustering: Although the gene clustering diagrams in Fig. 1 and 4 are commonly used, I had a difficult time in making sense of them in this context.

The IAC active module and the Network view are similarly very complicated and are not discussed in sufficient detail to enable a reasonable evaluation of the implications.

Discussion

The discussion is brief and could be expanded to include implications for the process of metastasis.

Reviewer #4 (Remarks to the Author):

I'm asked to comment on the proteomics part of the manuscript by Chee et al in which fibronectin-induced integrin adhesion complexes (IACs) isolated from two populations of metastatic cSCC cells were compared.

The described methods of mass spectrometry data acquisition and that of label-free quantitation are sound, but the reporting of the results is inadequate. A number of important details are missing:

- 1, how many peptides are required for a protein to be identified and qualified?
- 2, how many unique peptides are required?
- 3, how to handle a group of proteins sharing the same set of peptides (i.e. none of them has a unique peptide)? Are they reported as a group or only one protein is reported as a representative, or else? How is this representative selected?
- 4, what's the criterion for selecting differential regulated proteins (DEPs)?
- 5, how many proteins are there in each list of DEPs? How many of them and what percentage of them are core adhesion proteins?
- 6, For the t-test results reported in supplementary tables, which ones are based on Student's t- test and which ones on Welch's t-test?

Another comment, in Figure 1e and Supplementary Figure 1a, the color coding exaggerates the difference between Met-1 and Met-4. Figure 1 f is a more realistic depiction of the quantitation results.

Reviewers' comments:

Reviewer #1 (Remarks to the Author):

The Authors investigate the role of the protein called Mena in trans-nuclear membrane signaling. They find that Mena is enriched in IACs in cSCC cells and claim that it is connected to the LINC complex component Nesprin-2 with which it interacts and co-localizes at the nuclear envelope of metastatic cells. Mena seems to potentiate the interactions of Nesprin-2 with the actin cytoskeleton and the nuclear lamina and its depletion causes altered nuclear morphology, reduced tyrosine phosphorylation of the nuclear-membrane protein Emerin and downregulated expression of the immunomodulatory gene PTX3 via recruitment of its enhancer to the nuclear periphery. As such, the manuscript reveals novel connection between Mena and the LINC complex, it shows that Mena might regulate actin-nuclear lamina interactions and in this way affecting nuclear architecture and gene expression of cancer-relevant genes (especially those involved in cancer progression). In this way, it shows a direct link between proteins involved in adhesion and gene expression regulation, which might be disrupted or perturbed in cancer.

The study is very well presented and touches upon multiple very relevant topics such as metastasis, signal transduction and nuclear/chromatin organization. However, in its current form, the manuscript sends out too many messages which are not substantiated and provides a rather confusing picture, which would need to be framed into a more coherent one. Below I provide comments/suggestions which, if addressed, should strengthen the claims in the manuscripts.

Response

We thank the reviewer for their constructive comments and are pleased that they find our work very relevant and well presented. We have responded to all of the reviewer's comments and ensured that we deliver a more coherent message in the manuscript.

Major comments:

- The manuscript is very well written and clearly presents the data with a logical flow when it comes to individual figures. However, the big picture gets lost on the way and towards the end the reader can get confused by what the message really is. The authors introduce the manuscript by discussing the role of adhesion complexes in cancer progression and metastasis, yet this aspect is lost throughout the text aside from the fact that experiments are performed on Met4 cell line, which is derived from metastatic tumor. I would suggest either explaining the findings in the context of metastasis and how they specifically affect its onset or else stay clear from it altogether. On this note, it would be interesting to provide a more in-depth comparison with how Mena works in Met1 cell line, which is derived from the primary tumor (this is lacking completely).

Response and action taken

As the reviewer stated above, our study examined a number of novel cell biological aspects linked to Mena. In our revised manuscript, we have improved the framing of these interconnected elements in the text to enhance the flow of the paper. Since we identified

that Mena was enriched in adhesion complexes from Met4 cSCC cells, we used these cells as a tool to investigate Mena in our subsequent assays. Indeed, Mena appears to localise at the nuclear envelope in both the Met4 cell line and in another cSCC cell line, Met1 [Reviewer Fig. 1 below (not included in the manuscript)], suggesting that Mena is consistently recruited to the nuclear membrane in these tumour cells also, although relative quantification of SIM images is not possible. As we did not specifically address the role of Mena and the LINC complex in metastasis in this study, we have removed or toned down our comments about relevance to metastasis, as also requested by the editor. Some speculation about how our findings may relate to metastasis remains in the Discussion to highlight how future work may address links between our observations and cancer progression.

Reviewer Figure 1 2D-SIM imaging of Met1 and Met4 cSCC cells stained for lamin B1 (nuclear envelope marker) and Mena. Middle z-sections of cells are shown. Zoomed images (insets, right) are derived from the respective dashed white boxes. Scale bar, 2 μm (zoom, 0.5 μm).

- If the emphasis on metastasis is to remain in the manuscript I would suggest informing the reader about its mutation / expression status in various primary vs metastatic tumors.

Action taken

We have removed the emphasis on metastasis, as detailed above.

- The entire manuscript is based on the putative interaction between Mena and nesprin-2 and once this is established early on in the manuscript all the phenotypes the Authors observe upon Mena depletion are then explained by the loss of this interaction. The fact that Mena localizes throughout the cell and likely has other functions than the ones proposed by the Authors is ignored. It might very well be that the effects the Authors see (which are convincing on their own) stem from the effects of Mena depletion on the plasma membrane or/and the cytoskeleton, and I could easily imagine that if the cytoskeleton is

perturbed that will affect nuclear morphology. Therefore, the Authors should try to disentangle the effects Mena's interaction with nesprin-2 might have on the nuclear morphology (and related aspects), from the effects Mena might have when not in complex with nesprin-2. For this reason, the comparison of Mena depletion in Met1 and Met4 cells would be particularly useful. Related to this, would the depletion of nesprin-2 mimic the effects exerted by the depletion of Mena on the nucleus?

Response and action taken

To address this point, we first determined the polymerisation status of actin in the presence or absence of Mena by quantifying the proportion of F-actin to monomeric G-actin in these cells using biochemical fractionation. Knockout of Mena did not significantly affect the F-actin/G-actin ratio, indicating that the filamentous cytoskeleton was not perturbed in the absence of Mena (new **Supplementary Fig. 3c, d**). We also quantified total F-actin intensity by imaging phalloidin staining using confocal microscopy and found no significant changes upon Mena knockout (new **Supplementary Fig. 3e, f**), supporting the finding that Mena loss does not perturb global F-actin formation in these cells. Depletion of nesprin-2G has been shown previously not to grossly perturb organisation of the actin cytoskeleton (DOI: 10.1126/science.1189072). To assess the impact of Mena knockout on the organisation of the actin cytoskeleton at the nucleus, we quantified nucleus-proximal F-actin intensity by thresholding for phalloidin staining directly above the nucleus (nucleus-apical F-actin). We found a significant decrease in F-actin intensity above the nucleus of Mena-knockout cells (new **Supplementary Fig. 3g, h**), but no change in F-actin filament length or branching (new **Supplementary Fig. 3i**), suggesting that recruitment of actin to the nucleus – but not nuclear-proximal actin filament organisation – is regulated by Mena. Together, these additional data suggest that the observed effects of Mena loss are **not** as a result of disruption of cytoskeletal integrity per se, and they support our conclusion that Mena exerts its effects on the nucleus by regulating the coupling of F-actin to the nuclear envelope. We have included these data in new **Supplementary Fig. 3c–i** and described these details in the **Mena depletion causes LINC complex dissociation from actin and lamin A/C** subsection of the Results.

- The Authors should comment on the big picture of the changes they observe in actin-linked proteins (Fig 1f) or ECM or cell migration proteins (Fig. 4d) given that proteins belonging to all these groups get upregulated in both Met1 in comparison to Met4, Met4 in comparison to Met1, and in Mena wt with respect to Mena^{-/-} (and vice versa). I understand that the Authors chose one protein to focus on but since the manuscript addresses the question of metastasis and the role of the actin – lamina association in this process, seeing these results might be confusing without any guidance.

Response and action taken

The reviewer is correct that we detected multiple differences in adhesion, migration and actin-associated proteins/genes in our 'omics analyses. We had noted some of these specific protein changes in the text, but we now provide additional descriptions of the overall changes in the Results and Discussion text to better contextualise our findings. We have removed the emphasis on metastasis, as detailed above.

- The Authors have performed a lot of screens, have undertaken multiple high throughput approaches, yet from each approach they typically chose to focus on one selected candidate protein or a gene (as in the case of the DamID data and the *PTX3* gene). Is *PTX3* the only upregulated gene whose upregulation can be explained by the DamID data?

Response and action taken

We chose *PTX3* as an exemplar gene as it was the most down-regulated gene upon loss of Mena. We agree that it is useful to examine multiple candidates that emerged from our analyses, and we have now expanded the analysis of the DamID-seq data to include other highly dysregulated genes. We show that additional genes down-regulated in Mena-knockout cells are associated with increased recruitment of their enhancer regions to the nuclear lamina (associated with gene repression), consistent with our observations for *PTX3*. Conversely, genes up-regulated in Mena-knockout cells are associated with decreased lamina recruitment of their enhancer regions (associated with loss of gene repression), indicating an association between the direction of regulation of gene expression (up- or down-regulation) and the arrangement of gene regulatory regions of chromatin in the absence of Mena. These data extend our conclusions and imply that Mena regulates the expression of a set of cell migration and immune response genes by modulating the repositioning of their enhancers to the nuclear lamina. We present these data in a new **Supplementary Fig. 4g** and describe these details in the **Mena controls emerlin tyrosine phosphorylation and linked gene expression** subsection of the Results. In addition, we used network analysis to link *PTX3* to other Mena-dependent genes using reported co-expression and genetic interaction data (new **Fig. 4h**), further contextualising this set of highly dysregulated genes and suggesting another regulatory layer by which Mena could regulate the expression of these genes.

- [Fig 2a-b] The authors highlight the localization of Mena at the focal adhesion and at the nuclear membrane and claim co-localization between Nesprin-2 and Mena. This is not obvious from the image presented. Rather, the two signals are abutted. The Authors should be more accurate here. Given that the entire manuscript is built on the interaction between Mena and nesprin-2 I would recommend strengthening these claims perhaps by performing an even higher resolution imaging. What is most confusing though is that in the image shown it is not clear whether the proximity between Mena and actin occurs at the plasma membrane or at the nuclear membrane. Could the Authors indicate clearly where the plasma membrane would be in this image? Also, is the depletion of the Mena signal at the basal side of the nucleus/cell shown on this picture representative of all the cells? Could the Authors comment on it?

Response and action taken

Our analyses by confocal imaging showed co-localisation between Mena and nesprin-2 at the nuclear envelope, as indicated by black regions in the merged-channels zoom image (magenta + green = black) (Fig. 2a). We have now quantified the co-occurrence of Mena and nesprin-2 signals and show that they co-localise at the nucleus but not elsewhere in the cell (new **Fig. 2b**; see below). Super-resolution SIM imaging revealed this to be partial co-localisation at the nuclear envelope, as indicated by black regions in the merged-channels zoom image where the magenta and green signals abut (Fig. 2c). This partial overlap of

signal suggests that Mena and nesprin-2 may be in very close proximity or adjacent at the nuclear envelope, consistent with them forming a molecular complex (Fig. 2e, g, h). We had described this observation as “Mena co-localised at the apical-most region of nesprin-2 staining” in the text, but we have clarified this description in the **Mena and nesprin-2 form a complex and co-localise at the nuclear membrane** subsection of the Results as suggested by the reviewer. We have also added green arrowheads to **Fig. 2c** to indicate zones of Mena and nesprin-2 adjacency and partial overlap.

To further characterise the localisation of Mena at higher resolution, as recommended by the reviewer, we performed 3D stimulated emission depletion (STED) nanoscopy, which has approximately double the axial resolution of SIM (and better lateral resolution). Quantification of STED images from super-resolution *xz* scans showed overlapping Mena and nesprin-2 signals (new **Fig. 2d**), supporting our analyses by confocal and SIM imaging. Furthermore, staining of the plasma membrane using wheat germ agglutinin revealed that the co-localisation of Mena and nesprin-2 was set back from the plasma membrane. Mena–nesprin-2 co-staining partially overlapped with F-actin staining, which extended to the plasma membrane (new **Fig. 2d**). These new data imply that nuclear-proximal Mena localises at the nuclear envelope, where nesprin-2 resides, rather than the plasma membrane, while F-actin associates with both the plasma membrane and the nuclear envelope. We describe these findings in the **Mena and nesprin-2 form a complex and co-localise at the nuclear membrane** subsection of the Results, linked to new **Fig. 2d**.

We did not observe substantial depletion of Mena signal at the basal side of the nucleus in these cells. Orthogonal reconstruction of images acquired by super-resolution SIM showed Mena localisation at the basal nuclear membrane (**Reviewer Fig. 2** below). Moreover, *xz* scans acquired by super-resolution STED nanoscopy clearly show localisation of Mena around all sides of the nucleus (new **Fig. 2d**).

Reviewer Figure 2 3D-SIM imaging of Met4 cSCC cells stained for Mena, nesprin-2 and F-actin (detected using phalloidin). Orthogonal (*yz*) section of cell shown. Zoomed images (insets, bottom) are derived from the dashed white box. Scale bar, 3 μm (zoom, 1 μm).

- The model the Authors propose in Figure 5 points to massive and multiple effects caused by Mena removal: 1) detachment of the actin network entirely from the nuclear envelop, 2)

lack of Nesprin-2 – actin interaction, 3) lack of Nesprin-2 – Sun1/2 interaction, 4) lack of Sun1/2 with emerin interaction, 5) lack of emerin phosphorylation, 6) changes in the chromatin marks and 7) chromatin repositioning close to the nuclear membrane. To my understanding the Authors did not provide proofs for points 1, 3, 4 and commented only on one enhancer changing its proximity to the membrane (with regards to point 7). Hence, the model in my view is far-stretched. If there is a complete loss of connectivity between the actin cytoskeleton and the nuclear envelope I would expect even more drastic changes to the morphology of the cell and nucleus. In order to prove such drastic changes at the nuclear envelope EM images are recommended.

Response and action taken

We agree that some of the putative consequences of Mena loss illustrated in our working model are yet to be proved, so we have tempered the model to make it clear exactly which aspects of the nuclear mechanosensing mechanism have been demonstrated. The reviewer highlights four specific points in this comment for clarification (reviewer points 1, 3, 4 and 7). For reviewer point 1, we do not propose that the F-actin network is detached entirely from the nuclear envelope upon Mena removal, and we apologise if there was a lack of clarity here. Rather, more specifically, our findings indicate that there is disruption to the association of F-actin with the LINC complex component nesprin-2, which we show in **Fig. 3a** and new **Supplementary Fig. 3g, h**. We have modified **Fig. 5b** and the legend to Fig. 5 to reflect this more accurately. LINC complex interactions with actin can also be mediated by nesprin-1, which provides an alternative mode of connecting the actin cytoskeleton to the nucleus that does not involve nesprin-2. We therefore did not anticipate, nor do we observe (new **Supplementary Fig. 3g, h**), complete dissociation of perinuclear actin from the nucleus, and thus we would not necessarily expect more drastic changes to cellular and nuclear morphology than those that we found upon reduction of actin–nesprin-2 association (**Fig. 3c–f**). For reviewer points 3 and 4, we do not claim that these interactions are lost upon depletion of Mena, and we have modified **Fig. 5b** to reflect this. Indeed, we now show that Mena removal does not affect the ability of nesprin-2 to interact with the LINC complex component SUN2 (new **Supplementary Fig. 3a**), suggesting that the associations of the nuclear membrane proteins of the LINC complex are not disrupted upon loss of Mena, further clarifying the model. We have modified the title, Abstract and text to highlight the specific effects of Mena on nesprin-2, rather than the broader LINC complex, that we observe. For reviewer point 7, as discussed above, we have now expanded the analysis of the DamID-seq data to include a network of highly dysregulated genes in addition to the most down-regulated gene upon loss of Mena, *PTX3* (new **Fig. 4h** and new **Supplementary Fig. 4g**). These data support our conclusion that Mena regulates the expression of a set of cell migration and immune response genes by modulating the proximity of their enhancers to the nuclear membrane.

- Could the Authors provide information on how long do the cells survive after the depletion of Mena?

Response

The cells retained a stable knockout of Mena for multiple passages over three-month periods of culture (after which, fresh cells were resurrected from storage), with no impact on their survival during this time.

- The claims about the function of Mena at the nuclear membrane would be stronger if the Authors could show a direct interaction of Mena with nesprin-2 or other proteins of the complex. Also the microscopy study of the co-localization between Mena and Nesprin-2 would benefit from approaches such as PLA.

Response and action taken

We have not mapped direct interactions of Mena with nesprin-2, and guidance from the editor indicates that this is not required for this manuscript. As PLA is prone to false positives and is difficult to control adequately in our experience (DOI: 10.1101/411355), we have not used this method here. To strengthen our claims of the association between Mena and nesprin-2, we have used an alternative approach, STED nanoscopy, which confirmed and quantified the co-localisation of Mena and nesprin-2 at the nuclear envelope (new **Fig. 2d**; see above).

Minor Points:

- [line 163 and Fig 1 legend] The Authors use the following expression on a few occasions: “Actin-associated proteins significantly differentially regulated between Met1 and Met4 IACs”. I believe that what they refer to are simply differences in abundance, which do not say per se anything about differences in the regulation. I would recommend to use a more accurate description here.

Action taken

We have amended these phrases, as suggested by the reviewer, to refer specifically to differential protein abundance/enrichment in the text and the legend to Fig. 1.

- [line 346] The Pearson coefficient of 0.487 is not necessarily indicative of correlation. The Authors should add a word of caution, stating that the correlation they identified is very weak (borderline).

Action taken

We have amended the text accordingly.

- [line 378] I believe the authors meant “PTX3 enhancer” and not “PTX3”.

Action taken

We have corrected this text.

- [Fig 2b] No quantification of the Mena and Nesprin-2 co-localization is provided. Is the juxtaposition of the signal a result of Mena being present all over the cells? What is the enrichment score at the perinuclear region? How does this compare to the Met1 cells?

Response and action taken

We have now quantified the co-localisation of Mena and nesprin-2 signals as requested by the reviewer. We used Manders’ coefficient as a metric of co-localisation that is independent of signal proportionality and provides two comparators: the fraction of Mena

that co-localises with nesprin-2 (M_{Mena}) and the fraction of nesprin-2 that co-localises with Mena ($M_{\text{Nesprin-2}}$). This enables more effective quantification of co-localisation of proteins that also distribute to different locales in the cell, such as is the case for Mena and nesprin-2. We calculated Manders' coefficients for protein co-localisation at the nucleus and in the rest of the cell. These data, which we provide as a new **Fig. 2b**, show that Mena and nesprin-2 co-occur around the nucleus, but not elsewhere in the cell. This establishes a non-random pattern of co-localisation associated with the nucleus, further supporting our conclusion that Mena and nesprin-2 co-associate at the nuclear membrane.

We also quantified the co-localisation of Mena and nesprin-2 at the nucleus in Met1 cells as compared to Met4 cells. Co-occurrence analysis indicated some nuclear co-localisation of Mena and nesprin-2 in Met1 cells (**Reviewer Fig. 3** below), consistent with the recruitment of Mena to the nuclear membrane in these tumour cells also, as detailed above (**Reviewer Fig. 1** above). However, the fraction of Mena that co-localised with nesprin-2 at the nuclear membrane was significantly lower in Met1 cells as compared to Met4 cells, and the fraction of nesprin-2 that co-localised with Mena at the nucleus was slightly lower in Met1 cells (although this was not statistically significant) (**Reviewer Fig. 3** below). These data suggest that Mena localises to the nuclear membrane of both cSCC cell lines, but to a lesser extent in Met1 cells. As we did not set out to specifically address the role of Mena and the LINC complex in metastasis in this study, and as we have removed any inferences about relevance to metastasis, as also requested by the editor (as discussed above), we have not included these data in the manuscript.

Reviewer Figure 3 Nuclear co-occurrence analysis of Mena and nesprin-2 signals in Met1 and Met4 cells determined from spinning-disk confocal images. M_{Mena} , co-occurrence fraction of Mena with nesprin-2 at the nucleus in Met1 or Met4 cells (left panel). $M_{\text{Nesprin-2}}$, co-occurrence fraction of nesprin-2 with Mena at the nucleus in Met1 or Met4 cells (right panel). Black bar, mean; light grey box, range. * $P < 0.05$; n.s., not significant; two-sided Mann–Whitney U test for M_{Mena} , two-sided Student's t -test for $M_{\text{Nesprin-2}}$ ($n = 9$ and 14 images for Met1 and Met4 cells, respectively, from $n = 3$ independent experiments).

- [Fig 2c and Fig. 2e, f] Could the Authors comment on why in the nesprin-2 IP the two bands of Mena seem to be equally strong in the case of wt nesprin-2 (Fig 2c), while in the case of the truncations one of them seems much stronger (Fig 2e, f)?

Response

In Fig. 2e (formerly Fig. 2c), we immunoprecipitated endogenous nesprin-2 isoforms (using nesprin-2 antibody), which may contain additional binding sites for other variants of Mena. They also contain indirect Mena binding sites, such as the actin-binding CH domain at the N-terminus. As such, we may be pulling down the Mena variants, directly or indirectly, to a similar extent.

In Fig. 2g, h (formerly Fig. 2e, f), we pulled down truncated nesprin-2 with a GFP tag (using GFP-Trap). The truncated nesprin-2 may lack some Mena variants binding sites. In addition, GFP-nesprin-2G(SR49–56) (Fig. 2g) lacks the actin-binding CH domain, which will exclude indirect Mena variant binding via actin, whereas GFP-mini-nesprin-2GΔSR3–50 (Fig. 2h) has GFP adjacent to the CH domain at the N-terminal, which may hinder indirect interaction with other Mena variants via actin. We therefore expect some variability in Mena variants being pulled down.

- What is its expression status of PTX3 in Met1 cells (as well as its enhancer localization).

Response

As discussed above, we used Met4 cells as a tool to investigate Mena in this study as we did not set out to specifically address relevance to metastasis. Therefore, we did not investigate the expression status of *PTX3* or the positioning of its enhancer in Met1 cells. Rather than compare Met4 cells to a separate cell line (such as Met1), we used CRISPR/Cas9 in Met4 cells to generate cells with no nuclear Mena to compare to cells with high levels of nuclear Mena to better control our delineation of this biological mechanism.

- [Fig 3a-b] Given the high variability of the Met4 Mena^{-/+} Mena11a signals for both Actin and Lamin A/C it would be useful to also measure at the same time Mena in these pulled down complexes. Is there a reason why the Authors did not probe for Mena in these IP experiments? It would be interesting to see if the variability in the Actin or Lamin A/C signals were dependent on the Mena signal in the corresponding complexes.

Response

This would be interesting to assess; however, we could not probe the membranes for Mena as it has a similar molecular weight to lamin A/C, which would have obscured the bands on the membrane, and we kept the blotting analysis consistent for both sets of IPs.

- I suggest moving Fig 4a to Supplementary Figures.

Action taken

We have made this suggested move to a new **Supplementary Fig. 4a**.

Reviewer #2 (Remarks to the Author):

In their manuscript, Chee et al. investigated the systems-level changes in the integrin adhesome during the metastatic progression of a patient-derived cutaneous squamous cell carcinoma (cSCC). Their investigation revealed that the actin-regulator mammalian enabled

(mena) is enriched in integrin adhesion complexes (IACs) in metastatic cSCC cells. By computationally constructing a protein interaction network of the cSCC adhesome using curated physical protein-protein interactions from the BioGRID database, the authors found that mena was proximal in an actin-binding/regulation sub-network to the actin-binding outer nuclear membrane KASH protein nesprin-2. The authors went on to show that mena colocalizes with nesprin-2 at the nuclear envelope and the two proteins form a biochemical complex as demonstrated by co-immunoprecipitation and pull-down experiments. Chee et al. then provide evidence to suggest that mena is required for the ability of nesprin-2 to interact with the actin cytoskeleton as well as the nuclear lamina. In addition, mena knock-out (KO) cSCCs displayed altered nuclear morphology and nuclear mechanotransduction as well as down-regulated expression of the immunomodulatory PTX3 gene. Moreover, the recruitment of the PTX3 enhancer to the nuclear periphery was elevated in mena KO cSCCs relative to control cells. In summary, this work suggests that mena performs several unexpected, yet related, functions at the cytoplasm surface of the nuclear envelope where it appears to work together with nesprin-2-containing LINC complexes to control the physical coupling of the nucleus and the actin cytoskeleton, nuclear morphology, chromatin organization, and cancer gene expression. While the implications of this work are quite novel and exciting, I cannot recommend that it be accepted for publication in its current form due to the following major and minor issues:

Response

We thank the reviewer for their constructive comments and are encouraged that they find our work novel and exciting. We have responded to all of the reviewer's concerns below.

Major Issues:

1) The title states, "mena regulates the LINC complex". However, there is little evidence presented in this work to support this claim. As shown in Figure 5, LINC complexes are nuclear envelope-spanning molecular bridges composed of the outer and inner nuclear membrane KASH and SUN proteins, respectively. Yet, the authors neglect to address the potential impact of mena on either of the widely expressed mammalian SUN proteins: SUN1 and SUN2. For example, the authors should determine if mena-depletion impacts the ability of nesprin-2 to interact with SUN1 and/or SUN2 as well as the nuclear envelope localization of nesprin-2, SUN1, and SUN2.

Response and action taken

To determine if Mena depletion impacts the ability of nesprin-2 to interact with SUN proteins, as suggested by the reviewer, we immunoprecipitated endogenous nesprin-2 in Met4 cells and Met4 cells depleted of, and re-expressing, Mena and then probed for SUN2. Quantification of immunoblotting of nesprin-2-associated protein complexes revealed no statistically significant difference in SUN2 co-immunoprecipitation with nesprin-2 in the presence or absence of Mena. These data indicate that Mena does not regulate the interaction between nesprin-2 and SUN2 and suggest that removal of Mena does not disrupt the LINC complex biochemical complex. We have included these data in a new **Supplementary Fig. 3a**, described them in the **Mena depletion causes nesprin-2 dissociation from actin and lamin A/C** subsection of the Results and modified our working model in **Fig. 5b** to reflect this. We have also modified the title, Abstract and text to

highlight the specific effects of Mena regulating nesprin-2, rather than the integrity of the LINC complex, that we observe.

2) While the model provided by the authors in Figure 5A suggests that mena directly interacts with nesprin-2G at the outer nuclear envelope. Figure 5A also suggests that the authors know the part of mena that interacts with nesprin-2G. However, they do not present any data to support either suggestion. The authors really should determine if mena interacts directly with nesprin-2. They should then map the mean-binding site on nesprin-2 and vice versa. Armed with this information, the authors could then directly test the physiological relevance of the mena-nesprin-2 interaction by determining if a mena construct harboring a mutation that prevents its ability to interact with nesprin-2 can rescue the nuclear morphology and emerin tyrosin phosphorylation phenotypes observed in the mena KO cSCCs. This information is really important for two reasons: 1) The sub-network analysis presented in Figure 1H shows that mena likely interacts with nesprin-2 through hPrp4; and 2) nesprin-2 was not identified in the meta-adhesome reported by Horton et al. (2015), while nesprin-1, nesprin-3, SUN1, and SUN2 were.

Response and action taken

While we have determined that Mena and nesprin-2 are in the same biochemical complex and co-occur around the nucleus (including new experiments and analyses in **Fig. 2b, d**), we have not mapped direct interactions of Mena with nesprin-2, as per guidance from the editor. We have modified our model in **Fig. 5a** and the legend to Fig. 5 to indicate that the interaction region of Mena has not been defined.

With regard to the cSCC adhesome sub-network analysis, the reviewer is correct that our analysis reveals known associations that exist between hPrp4 and both Mena and nesprin-2, as presented in **Fig. 1i** (formerly Fig. 1h). We probed for hPrp4 in nesprin-2 co-immunoprecipitation experiments but its molecular weight of ~58 kDa resulted in its band migrating too closely to that of IgG heavy chain by SDS-PAGE to enable interpretation, preventing detection of hPrp4 by immunoblotting. However, lack of reported evidence for an interaction between Mena and nesprin-2 in the BioGRID database (from which our networks were constructed) is not evidence of absence of an association in certain cell types, subcellular locations or experimental conditions. Therefore, while it remains to be determined if hPrp4 interacts with Mena and nesprin-2 in Met4 cells, our evidence implies that Mena and nesprin-2 form a biochemical complex.

With regard to the meta-adhesome database, this was derived from integrin adhesion complexes isolated from chiefly fibroblast cell lines, as well a malignant melanoma and a chronic myelogenous leukaemia cell line. The keratinocyte cell lineage was not represented in the meta-adhesome, so it is possible that some epidermal keratinocyte adhesion complex proteins are not described in the database. **Thus, our cSCC adhesome analysis here represents, to our knowledge, the first proteomic characterisation of keratinocyte-derived integrin adhesion complexes.** Moreover, there is evidence for the localisation of nesprin-2 at sites of cell adhesion in U2OS osteosarcoma cells: CH domain-containing isoforms of nesprin-2 have been detected in focal adhesions (DOI: 10.1371/journal.pone.0040098) and at cell-cell junctions in conditions of high calcium (DOI: 10.1016/j.yexcr.2016.06.008),

suggesting that localisation of nesprin-2 to integrin (and cadherin) adhesion complexes may be restricted to certain cell types.

3) Ena/VASP proteins promote actin filament assembly at several subcellular locations by acting as processive actin polymerases. Yet, the authors of this work seem to be proposing that mena controls the coupling of the nucleus with the actin cytoskeleton solely by promoting the ability of nesprin-2G to interact with perinuclear actin filaments. If this were true, I would anticipate that the re-expression of a mena construct harboring a mutation that disables its polymerase activity without impacting its ability to interact with actin in the mena-KO Met4 cells would rescue their nuclear morphology and emerin tyrosin phosphorylation phenotypes. In addition, it would be good for the authors to quantitatively determine if the absence of mena has an impact on the organization of perinuclear actin in the Met4 cells.

Response and action taken

To address this point (as also discussed in response to Reviewer 1, above), we first determined the polymerisation status of actin in the presence or absence of Mena by quantifying the proportion of F-actin to monomeric G-actin in the Met4 cells using biochemical fractionation. Knockout of Mena did not significantly affect the F-actin/G-actin ratio, indicating that the filamentous cytoskeleton was not perturbed in the absence of Mena (new **Supplementary Fig. 3c, d**). We also quantified total F-actin intensity by imaging phalloidin staining using confocal microscopy and found no significant changes upon Mena knockout (new **Supplementary Fig. 3e, f**), supporting the finding that Mena loss does not perturb global F-actin formation in these cells. Results from mouse skin melanoma cells showed that the Mena paralogue VASP (also expressed in our cSCC cells) is able to compensate for loss of Mena and Evl, the other mammalian Ena/VASP family protein, which otherwise depletes F-actin (DOI: 10.7554/eLife.55351), so our data are consistent with the overlapping functions of Ena/VASP proteins, and/or other proteins with polymerase activity, such as WASP family proteins (DOI: 10.15252/embj.201797039), providing potential redundancy to maintain actin polymerisation. Thus, Mena is not essential for actin filament assembly in Met4 cells.

To assess the impact of Mena loss on the organisation of the actin cytoskeleton at the nucleus, we next quantified nucleus-proximal F-actin intensity by thresholding for phalloidin staining directly above the nucleus (nucleus-apical F-actin). We found a significant decrease in F-actin intensity above the nucleus of Mena-knockout cells (new **Supplementary Fig. 3g, h**), but no change in F-actin filament length or branching (new **Supplementary Fig. 3i**), suggesting that capture of actin at the nucleus – but not nuclear-proximal actin filament organisation – is regulated by Mena. Together, these additional data suggest that the observed effects of Mena are not as a result of its promotion of actin polymerisation, and they support our conclusion that Mena exerts its effects on the nucleus by regulating the coupling of F-actin to the nuclear envelope. We have included these data in new **Supplementary Fig. 3c–i** and described these details in the **Mena depletion causes LINC complex dissociation from actin and lamin A/C** subsection of the Results.

4) In Figure 3A and 3B, the authors show the results of co-immunoprecipitation experiments suggesting that the interaction between nesprin-2 and actin or A-type lamins is disrupted in

the absence of mena. While these results appear to be compelling, I would strongly caution the authors against drawing too many conclusions from them without support from orthogonal experimental evidence (i.e. FRET, FRAP, or co-localization). The presence of actin or A-type lamins in a co-immunoprecipitation should be concerning due to their ability to form mesh-like polymers that may non-specifically trap proteins resulting in them being co-immunoprecipitated. It should also be said that the effect of mena-depletion on the nesprin-2-A-type lamin interaction could either be due to mena impacting the ability of nesprin-2 to directly interact with SUN1/2 or A-type lamins. Given these results, are the authors suggesting that the interaction of nesprin-2G with mena induces an allosteric conformational change that modulates its ability to interact with either actin or A-type lamins? If so, they need to be more explicit.

Response and action taken

To provide additional experimental evidence of the Mena-dependent association of actin with the nuclear envelope, we quantified nucleus-proximal F-actin intensities directly above the nucleus (nucleus-apical F-actin, as described above). We found a significant decrease in F-actin intensity above the nucleus of Mena-knockout cells (new **Supplementary Fig. 3g, h**), but not total cellular F-actin intensity (new **Supplementary Fig. 3e, f**) nor nucleus-apical F-actin organisation (new **Supplementary Fig. 3i**), supporting our conclusion that Mena regulates the recruitment of actin to the nucleus.

To determine if Mena depletion impacts the ability of nesprin-2 to interact with SUN proteins, we immunoprecipitated endogenous nesprin-2 in Met4 cells and Met4 cells depleted of, and re-expressing, Mena and then probed for SUN2 (as described above). Quantification of immunoblotting of nesprin-2-associated protein complexes revealed no statistically significant difference in SUN2 co-immunoprecipitation with nesprin-2 in the presence or absence of Mena. These data indicate that Mena does not regulate the interaction between nesprin-2 and SUN2. We have included these data in a new **Supplementary Fig. 3a**, added clarification to the **Mena depletion causes nesprin-2 dissociation from actin and lamin A/C** subsection of the Results and modified our working model in **Fig. 5b** to reflect this. We have not performed conformational dynamics or structural studies of nesprin-2, and we do not demonstrate herein that Mena is an allosteric effector of nesprin-2, but this is a very interesting idea and we have raised this possibility in the Discussion and will address it beyond this paper.

5) In Figure 2a the nuclear envelope localization of mena in Met4 cells grown in 2D is clear, while this localization is apparently lost when Met4 cells are grown in 3D as shown in Figure 3C. Moreover, it appears that the nuclear envelope localization of mena in 2D cultured Met4 cells correlates with the levels of nesprin-2 present within the nuclear envelope. Based on these conflicting results, it is unclear to me how mena is being recruited to the nuclear envelope in the sCC cells in a nesprin-2-dependent manner. Therefore, does mena lose its ability to localize to the nuclear envelope in 3D cultured cells because nesprin-2 no longer is found at the nuclear envelope in cells grown under these conditions? It would also be good to know if disrupting the nuclear envelope localization of nesprin-2 by either knocking-down nesprin-2 or over-expressing a LINC complex-disrupting dominant negative construct (e.g. the KASH domain of nesprin-2) displaces mena from the nuclear

envelope.

This information would be critical for the authors to be able to distinguish between mena working through nesprin-2G-containing LINC complexes or focal adhesions.

Response and action taken

The nuclear envelope localisation of Mena is not lost when Met4 cells are grown in 3D. The images in Fig. 3c are 3D reconstructions of cells grown in 3D matrix, but when we examine individual z-slices from these reconstructions, we observe localisation of Mena around the nuclear periphery, which was not as clear in the 3D reconstruction images that we used to quantify nuclear volume. We have therefore included these additional images in a new **Supplementary Fig. 3b** and referred to this in the **Mena depletion causes nesprin-2 dissociation from actin and lamin A/C** subsection of the Results to clarify that Mena localises at the nucleus also in cells grown in 3D matrix.

6) In Figure 3C, the authors show that the volume of nuclei in mena-KO Met4 cells is significantly reduced relative to control cells. In Figure 3D, the authors show that nuclear height is significantly increased in mena-KO cells relative to Met4 cells expressing endogenous mena or mena-KO cells expressing GFP-mena11a. However, the F-actin images shown in Figure 3C suggest that the cytoplasmic area and volume of mena-KO Met4 cells may be reduced relative to control cells. Since nuclear and cytoplasmic volumes scale linearly in cells, the authors need to also quantify the effect of mena-KO on cytoplasmic volume. This is particularly important given the established role for mena as a regulator of actin polymerization, which is critical for cellular spreading and nuclear height control.

Response

While this would be interesting to analyse further, guidance from the editor indicates that quantification of the effects of Mena knockout on cell/cytoplasmic volume is not required for this re-submission. However, we do show that Mena is not essential for actin polymerisation or organisation in these cells (new **Supplementary Fig. 3c–i**; see above) and that Mena-knockout cells appear to spread on 2D matrix to a similar extent to parental cells (new **Supplementary Fig. 3e**). The morphology changes noted for Fig. 3c may, therefore, be a feature of cells migrating through 3D matrix, and we will follow this observation up separately.

7) While the dam-ID results presented in Figure 4 are compelling, the authors need to provide support from orthogonal experimental evidence for the repositioning of the PTX3 enhancer within the nucleoplasm (e.g. FISH or dCas9-mediated imaging of genomic loci). Moreover, the authors should show if re-expression of GFP-mena11a in their mena-KO cells rescues the phenotypes reported in Figures 4G-H. Since they have the GFP-mena11a-expressing mena-KO cells (see Figs. 3A-B), this request should not be too difficult. Moreover, it would be good to know if the expression levels of PTX3 are also decreased in cells lacking nesprin-2 or expressing a LINC complex-inhibiting dominant negative construct.

Response and action taken

To provide further support for the repositioning of the *PTX3* enhancer region identified by lamin B1 DamID-seq (Fig. 4e) and independently verified by DamID-qPCR (Fig. 4f), we

performed emerin ChIP-qPCR, alongside an IgG ChIP control, to quantify the association of the *PTX3* enhancer region with the inner nuclear membrane protein emerin. Using two different ChIP-qPCR protocols (presented separately), we observed increased association of *PTX3* with emerin in *Mena*^{-/-} cells, consistent with its repositioning to the nuclear lamina in the absence of Mena as shown by DamID-seq and DamID-qPCR (**Reviewer Fig. 4** below). Owing to the close proximity of the background region to the region of interest in the ChIP-qPCR experiments, however, these data were not normalised to the background region and were presented as percentage input versus IgG controls. We therefore do not include these data in the manuscript but present them here for the information of the reviewer.

Reviewer Figure 4 Nuclear lamina association of *PTX3* quantified by two different emerin ChIP-qPCR protocols (methods 1 and 2). qPCR primers amplified the *PTX3* enhancer region identified by lamin B1 DamID-seq.

We analysed the recruitment of chromatin to the nuclear periphery in cells re-expressing GFP-Mena11a, alongside parental Met4 cells and Met4 *Mena*^{-/-} cells, using lamin B1 DamID-seq. We found that re-expression of Mena resulted in a loss of association of the *PTX3* enhancer region with the nuclear lamina, indicating rescue of chromatin repositioning (**Reviewer Fig. 5a** below). However, quantification of *PTX3* expression in these cells using RT-qPCR did not reveal a reversion of *PTX3* expression to levels observed in the parental Met4 cells (**Reviewer Fig. 5b**). This could be explained by the potential retention of a repressed state of *PTX3*, despite relocation of the enhancer region away from the nuclear lamina, which has been observed previously for other genes (DOIs: [10.1016/j.cell.2013.02.028](https://doi.org/10.1016/j.cell.2013.02.028); [10.1126/science.1259587](https://doi.org/10.1126/science.1259587)). Thus, as considerable further work is required to resolve this phenomenon, we do not include these data in the manuscript and will pursue these experiments separately.

Reviewer Figure 5 Rescue of chromatin repositioning by re-expression of Mena accompanied by potential retention of *PTX3* repression. **(a)** Lamin B1 DamID-seq tracks were generated from Met4 (red, top profile), Met4 Mena^{-/-} (orange, middle profile) and Met4 Mena^{-/-} +GFP-Mena11a (blue, bottom profile) cells. The highest-scoring putative enhancer region associated with *PTX3* is indicated with a dashed grey box. **(b)** Quantification of *PTX3* expression in Met4, Met4 Mena^{-/-} and Met4 Mena^{-/-} +GFP-Mena11a cells by RT-qPCR. Gene expression was normalised to *GAPDH* and expressed relative to Met4 cells. **** $P < 0.0001$; two-sided Student's *t*-test ($n = 3$ independent experiments).

8) Why do the authors only report the effect of deleting mena from Met4 cells on the positioning of *PTX3*'s enhancer relative to lamin B1? Do the enhancers of the other genes that are up-regulated in Met4 cells behave similarly to the *PTX3* enhancer? Conversely, what happens to the enhancers of the genes that are up-regulated in the mena-KO Met4 cells?

Response and action taken

We chose *PTX3* as an exemplar gene as it was the most down-regulated gene upon loss of Mena (as also discussed in response to Reviewer 1, above). We agree that it is useful to examine multiple candidates that emerged from our analyses, and we have now expanded the analysis of the DamID-seq data to include other highly dysregulated genes. We show that additional genes down-regulated in Mena-knockout cells are associated with increased recruitment of their enhancer regions to the nuclear lamina (associated with gene repression), consistent with our observations for *PTX3*. Conversely, genes up-regulated in Mena-knockout cells are associated with decreased lamina recruitment of their enhancer regions (associated with loss of gene repression), indicating an association between the direction of regulation of gene expression (up- or down-regulation) and the arrangement of gene regulatory regions of chromatin in the absence of Mena. We present these data in a new **Supplementary Fig. 4g** and describe these details in the **Mena controls emerin tyrosine phosphorylation and linked gene expression** subsection of the Results. In addition, we used network analysis to link *PTX3* to other Mena-dependent genes using reported co-expression and genetic interaction data (new **Fig. 4h**), further contextualising this set of highly dysregulated genes and suggesting another regulatory layer by which Mena could regulate the expression of these genes. Together, these data extend our conclusions and imply that Mena regulates the expression of a set of cell migration and immune response genes by modulating the repositioning of their enhancers to the nuclear lamina.

Minor Issues:

1) Human gene names are written in all caps and italics.

Action taken

All human gene names have been written as stated by the reviewer.

2) Page 4, Line 93: The authors state that nesprin-1, -2, -3, and -4 make up “the outer nuclear membrane nesprin family proteins”. However, the authors should also include KASH5 and LRMP in this list.

Action taken

This addition has been made to the text.

3) The drawings of the GFP-nesprin-2G SR49-56 and Δ SR3-50 constructs shown in Figure 2D seem to have large stretches of light purple that are not present in the drawing of full-length nesprin-2G shown above. What are these light purple stretches?

Action taken

The construct diagram of GFP-nesprin-2G(SR49–56) has been adjusted to include indication of only the relevant domains.

4) The labels of Figures 2E and 2F should reflect the names of the GFP-tagged nesprin-2G constructs used to perform the pull-down experiments.

Action taken

The figure panel labels have been modified as suggested by the reviewer.

5) The name of the GFP-nesprin-2G Δ SR3-50 should be changed to GFP-mini-nesprin-2G Δ SR3-50, which is its original name per Jayo et al., 2016 Dev Cell where it was described.

Action taken

The name has been changed in the figures and text accordingly.

6) I do not believe that the statement “mena localizes immediately adjacent to nesprin-2” (page 4, line 114) can be made without quantification or with the ~300 nm axial resolution afforded by Nikon SIM under optimal conditions.

Response and action taken

To further characterise the localisation of Mena at higher resolution, we performed 3D stimulated emission depletion (STED) nanoscopy (see also comments to Reviewer 1, above), which has approximately double the axial resolution of SIM (and better lateral resolution). Quantification of STED images from super-resolution xz scans showed overlapping Mena and nesprin-2 signals (which we provide as a new **Fig. 2d**), supporting our analyses by confocal and SIM imaging and our statement in the text.

7) Please identify in the Materials and Methods section the fluorescence filters used in the authors' Dragonfly and Nikon SIM imaging systems.

Action taken

Details of filters used have now been included in the Materials and Methods section.

Reviewer #3 (Remarks to the Author):

The authors compare two BSCC variants from the same patient; one from the initial tumor (Met1) and the second from a metastatic tumor in a lymph node (Met4). Of the many proteins that are different between the two lines, they focus on Mena which is upregulated in the metastatic Met4 line. Mena interacts with Nesprin 2 and localizes to the nuclear membrane. There are changes in the expression profiles, nuclear shape, emerin phosphorylation and H3 methylation with the loss of Mena. The exact mechanisms of these changes are not detailed but a lot of circumstantial evidence is provided. Although I don't feel comfortable in evaluating the nodes in their modeling, there is clearly an important role for Mena. Missing are cell migration assays and direct measurements of metastasis; however, this is a lot to expect in an already data heavy paper. The major message of this paper is that Mena interacts with Nesprin 2 and their interaction is implicated in the nuclear morphology and expression pattern changes with Mena depletion. This is an important set of findings even though the cell biological implications of the changes have not been fully explored. Because of the large amount of data that may be useful to other labs, I recommend publication with only minor revisions.

Response

We thank the reviewer for their evaluation. We are encouraged that only minor issues were raised, which we have addressed below.

Results

Gene clustering: Although the gene clustering diagrams in Fig. 1 and 4 are commonly used, I had a difficult time in making sense of them in this context.

Action taken

To improve the clarity of the clustered heatmaps in **Figs 1 and 4**, we have added a key to indicate the "axes" of each heatmap, we have annotated the colour scale bar to aid interpretation of the colours in the heatmap gradient, and we have demarcated the functional annotation (orange/blue cells) with a black box to separate these from the protein/gene quantification heatmap (green/white/purple cells).

The IAC active module and the Network view are similarly very complicated and are not discussed in sufficient detail to enable a reasonable evaluation of the implications.

Action taken

To enable better evaluation of the network analyses, we have added more detailed description of the rationale behind our network biology approach, the approach itself and the implications of the findings to the **Characterisation of the adhesome of cSCC** subsection of the Results. To provide further clarity, we have added a workflow describing steps in the network analysis approach to a new **Fig. 1g** and a key explaining the visualisation of the active module as a new **Supplementary Fig. 1g**.

Discussion

The discussion is brief and could be expanded to include implications for the process of metastasis.

Response and action taken

We have removed any reference to how what we have observed here relates causally to the metastatic process (as discussed in response to Reviewer 1, above). However, we have added to the Discussion by including consideration of new results now included in response to reviewers' comments.

Reviewer #4 (Remarks to the Author):

I'm asked to comment on the proteomics part of the manuscript by Chee et al in which fibronectin-induced integrin adhesion complexes (IACs) isolated from two populations of metastatic cSCC cells were compared.

The described methods of mass spectrometry data acquisition and that of label-free quantitation are sound, but the reporting of the results is inadequate. A number of important details are missing:

Response

We thank the reviewer for assessing the proteomic aspects of the manuscript. We have added the requested methodological details to the text and supplementary information, as summarised below.

1, how many peptides are required for a protein to be identified and qualified?

Response and action taken

At least two peptides were required for a protein to be identified. PSM and protein FDRs were maintained at 0.01. At least one peptide ratio was required for label-free quantification. We have added the missing minimum peptide information to the **MS data analysis** subsection of the Methods.

2, how many unique peptides are required?

Response and action taken

At least one unique or razor peptide was required for protein identification. We have added this detail to the **MS data analysis** subsection of the Methods.

3, how to handle a group of proteins sharing the same set of peptides (i.e. none of them has a unique peptide)? Are they reported as a group or only one protein is reported as a representative, or else? How is this representative selected?

Response and action taken

The first protein accession of a group of proteins sharing the same set of peptides is reported as a representative. We have added this detail to the **MS data analysis** subsection of the Methods. In addition, to facilitate assessment of protein groups sharing peptides, we have expanded **Supplementary Table 1** to include protein accessions and peptide counts for all respective proteins in a protein group, which also remains available in the MS search data deposited in ProteomeXchange.

4, what's the criterion for selecting differential regulated proteins (DEPs)?

Response and action taken

Proteins enriched by at least two-fold with an FDR-corrected P value less than 0.05 ($q < 0.05$) were considered significantly differentially enriched. We have added this information to the **MS data analysis** subsection of the Methods and to the **Characterisation of the adhesome of cSCC** subsection of the Results.

5, how many proteins are there in each list of DEPs? How many of them and what percentage of them are core adhesion proteins?

Response and action taken

Of the identified proteins, 402 (23.3%) were differentially regulated, with 238 enriched in Met4 IACs and 164 enriched in Met1 IACs. Of the differentially regulated IAC proteins, 10 (2.49%) were core cSCC adhesion proteins (which represents 17.9% of the 56 identified core cSCC adhesion proteins). We have added these details to the **Characterisation of the adhesome of cSCC** subsection of the Results, and we have extended **Supplementary Table 1** to indicate all differentially regulated proteins.

6, For the t-test results reported in supplementary tables, which ones are based on Student's t- test and which ones on Welch's t-test?

Response

The respective method used for each t-test has been added as a new column in **Supplementary Tables 1–4**.

Another comment, in Figure 1e and Supplementary Figure 1a, the color coding exaggerates the difference between Met-1 and Met-4. Figure 1 f is a more realistic depiction of the quantitation results.

Action taken

We have modified the colour gradients used to visualise the data in heatmaps in **Fig. 1e** and **Supplementary Fig. 1a** by extending the range of the gradient scales.

REVIEWERS' COMMENTS

Reviewer #1 (Remarks to the Author):

I thank the Authors for addressing all of my comments extremely thoroughly. I find the manuscript much improved and I highly recommend it for publication by Nature Communications.

Reviewer #2 (Remarks to the Author):

Overall, I feel that the authors have successfully addressed the majority of my concerns. I recommend this manuscript for publication.

Reviewer #4 (Remarks to the Author):

The authors have addressed adequately all of my concerns. The manuscript has now passed my proteomics data quality check.

REVIEWERS' COMMENTS

Reviewer #1 (Remarks to the Author):

I thank the Authors for addressing all of my comments extremely thoroughly. I find the manuscript much improved and I highly recommend it for publication by Nature Communications.

Response

We thank the reviewer for their comments and for highly recommending publication of the manuscript in *Nature Communications*.

Reviewer #2 (Remarks to the Author):

Overall, I feel that the authors have successfully addressed the majority of my concerns. I recommend this manuscript for publication.

Response

We are very pleased that the reviewer recommends the manuscript for publication, and we are grateful for their constructive feedback during the review process.

Reviewer #4 (Remarks to the Author):

The authors have addressed adequately all of my concerns. The manuscript has now passed my proteomics data quality check.

Response

We thank the reviewer for their assessment of the manuscript and are pleased that they are satisfied with our revisions.